EMBO
Molecular Medicine

# The ASC inflammasome adapter governs SAA-derived protein aggregation in inflammatory amyloidosis

Marco Losa [1], Marc Emmenegger [1], Pierre De Rossi[2,11], Patrick M Schürch [3,11], Tetiana Serdiuk[4,11],
Niccolò Pengo[5], Danaëlle Capron [5], Dimitri Bieli[5], Niklas Bargenda [2], Niels J Rupp[6,7],
Manfredi C Carta[1], Karl J Frontzek[1], Veronika Lysenko[3], Regina R Reimann[1], Petra Schwarz [1],
Mario Nuvolone [1,8], Gunilla T Westermark[9], K Peter R Nilsson[10], Magdalini Polymenidou [2],
Alexandre PA Theocharides[3], Simone Hornemann [1], Paola Picotti [4] & Adriano Aguzzi [1✉]

## Abstract

Extracellularly released molecular inflammasome assemblies -ASC specks- cross-seed Aβ amyloid in Alzheimer's disease. Here we show that ASC governs the extent of inflammation-induced amyloid A (AA) amyloidosis, a systemic disease caused by the aggregation and peripheral deposition of the acute-phase reactant serum amyloid A (SAA) in chronic inflammatory conditions. Using super-resolution microscopy, we found that ASC colocalized tightly with SAA in human AA amyloidosis. Recombinant ASC specks accelerated SAA fibril formation and mass spectrometry after limited proteolysis showed that ASC interacts with SAA via its pyrin domain (PYD). In a murine model of inflammatory AA amyloidosis, splenic amyloid load was conspicuously decreased in $Pycard^{-/-}$ mice which lack ASC. Treatment with anti-ASC[PYD] antibodies decreased amyloid loads in wild-type mice suffering from AA amyloidosis. The prevalence of natural anti-ASC IgG ($-\log EC_{50} \geq 2$) in 19,334 hospital patients was <0.01%, suggesting that anti-ASC antibody treatment modalities would not be confounded by natural autoimmunity. These findings expand the role played by ASC and IL-1 independent inflammasome employments to extraneural proteinopathies and suggest that anti-ASC immunotherapy may contribute to resolving such diseases.

**Keywords** Innate Immunity; ASC; Serum Amyloid A (SAA); Inflammation; Amyloidosis
**Subject Category** Immunology

See also: S Maasewerd & BS Franklin

## Introduction

Inflammation-associated amyloid A (AA) amyloidosis occurs in a heterogeneous spectrum of chronic conditions, including inflammatory bowel disease, tuberculosis, hepatitis, genetic inflammatory diseases (e.g., familial Mediterranean fever), cancer as well as autoimmune diseases such as rheumatoid arthritis and vasculitis (Brunger et al, 2020; Lee et al, 2020; Papa and Lachmann, 2018). In these conditions, cytokines stimulate hepatocytes to synthesize and secrete Serum Amyloid A (SAA) into the bloodstream. During the acute-phase response, serum SAA can increase 1000-fold from its baseline concentration (Sack, 2018; Ye and Sun, 2015). Persistently high levels of SAA in serum can hamper its proper processing and clearance, leading to nucleation of aggregated AA fibrils and systemic deposition of AA amyloid. The deposition of amyloid in the spleen, kidney, liver, and heart can be massive and cause life-threatening disruption of tissue integrity (Chamling et al, 2021; Dubrey et al, 1996; Westermark and Westermark, 2009).

There is increasing evidence for an important role of innate immunity in the pathogenesis of protein misfolding diseases (PMDs) (Aguzzi, 2022; Anders and Muruve, 2011; Heneka et al, 2015; Heneka et al, 2014; Jang et al, 2019). The adapter protein ASC (Apoptosis-associated speck-like protein containing a caspase recruitment domain) plays an eminent role in the pathogenesis of Alzheimer's disease (AD) (Dansokho and Heneka, 2018; Ising et al, 2019; Venegas et al, 2017). Intrahippocampal injection of microglia-derived ASC specks leads to amyloid β (Aβ) cross-seeding in mice overexpressing Amyloid-β Precursor Protein and Presenilin-1 (APP/PS1;$Pycard^{+/+}$ mice) which is reduced in APP/

[1]Institute of Neuropathology, University Hospital Zurich, Zurich, Switzerland. [2]Department of Quantitative Biomedicine, University of Zürich, Zurich, Switzerland. [3]Department of Medical Oncology and Hematology, University Hospital Zurich, Zurich, Switzerland. [4]Institute of Molecular Systems Biology, Department of Biology, ETH Zurich, Zurich, Switzerland. [5]Mabylon AG, Schlieren, Zurich, Switzerland. [6]Department of Pathology and Molecular Pathology, University Hospital Zurich, Zurich, Switzerland. [7]Faculty of Medicine, University of Zurich, Zurich, Switzerland. [8]Amyloidosis Research and Treatment Center, Fondazione Istituto di Ricovero e Cura a Carattere Scientifico (IRCCS) Policlinico San Matteo, University of Pavia, Pavia, Italy. [9]Department of Medical Cell Biology, Uppsala University, Uppsala, Sweden. [10]Department of Physics, Chemistry and Biology, Linköping University, Linköping, Sweden. [11]These authors contributed equally: Pierre De Rossi, Patrick M Schürch, Tetiana Serdiuk. ✉E-mail: adriano.aguzzi@usz.ch

PS1;*Pycard*$^{-/-}$ mice. This increase in Aβ pathology can be prevented by co-injection of ASC specks with anti-ASC antibodies (Fernandes-Alnemri and Alnemri, 2008; Venegas et al, 2017).

Aβ and AA amyloid consist of β-pleated sheets and have similar fibril sizes, fibril strand orientations, and cross-seeding capability (Eisenberg and Jucker, 2012). Indirect evidence suggests a role for ASC in the pathogenesis of inflammation-associated (AA) amyloidosis. ASC was shown to co-localize with AA amyloid in kidney biopsies from patients with AA amyloidosis secondary to familial Mediterranean fever, where a gain-of-function mutation in the *Pyrin* gene activates ASC inflammasomes and leads to chronic overexpression of SAA (Balci-Peynircioglu et al, 2008). Also, cryopyrin-associated periodic syndrome (CAPS) patients, owing to a genetic condition, have constitutively high NLRP3 and ASC inflammasome activation (Pastore et al, 2013; Scarpioni et al, 2016) and are at high risk of developing AA amyloidosis (Brunger et al, 2020). SAA activates the NLRP3 inflammasome of human myeloid cells via an interaction with the P2X7 receptor, which induces ASC release in vitro through a cathepsin B-sensitive pathway (Niemi et al, 2011). In addition, the SAA cascade may prompt the upregulation of pro-inflammatory cytokines and enhances fibrillogenic activity toward SAA1 in vitro in murine macrophages (Chen et al, 2020; Gaiser et al, 2021).

In a murine model of AA amyloidosis, subcutaneous administration of silver nitrate (AgNO$_3$), coupled with an intravenous application of preformed SAA fibrils (known as Amyloid-Enhancing Factor or AEF), rapidly induces AA amyloidosis in spleen (Kisilevsky and Boudreau, 1983; Lundmark et al, 2013; Sponarova et al, 2013). AgNO$_3$ serves as a pro-inflammatory stimulus that leads to elevation in SAA serum levels, while preformed AA fibrils (AEF) serve as a template for SAA aggregation (Magy et al, 2003). Interestingly, depletion of splenic macrophages delays or inhibits AA amyloid accumulation in mice (Kennel et al, 2014; Lundmark et al, 2013) and macrophages can clear AA amyloid via Fc-receptor-mediated phagocytosis (Nyström and Westermark, 2012). These results suggest that components of the innate immune system may control, both positively and negatively, the course of AA amyloidosis.

Here, we investigated the role of ASC in inflammation-associated amyloidosis in vitro and in vivo. We found that ASC forms complexes with SAA in post-mortem tissue of a patient with inflammation-associated AA amyloidosis, colocalizes with murine splenic AA amyloid, accelerates SAA fibril formation, and interacts with SAA via its pyrin domain. *Pycard*$^{+/+}$ mice exhibited progressive splenic amyloidosis and a concomitant decrease in SAA serum concentration, both of which were attenuated in the absence of ASC. Treatment with anti-ASC antibodies decreased amyloid loads and improved health in a mouse model of amyloidosis, suggesting that anti-ASC immunotherapy may be useful in this condition.

## Results

### Colocalization of ASC with amyloid A

As ASC colocalizes with Aβ amyloid in mice and humans (Venegas et al, 2017), we asked whether ASC would co-localize also with AA, a type of extraneural amyloid composed of SAA fibrils. We used the amyloidotropic dye Congo Red (CR) and antibodies against ASC to

stain cardiac tissues from a patient suffering from vasculitis and systemic inflammation-associated AA amyloidosis and, for control, a 68-year-old male patient without any clinical sign of amyloidosis who died from left ventricular failure after myocardial infarction (Fig. 1A). CR-stained amyloidotic tissue, but not control tissue, showed a characteristic red appearance (Fig. 1B). ASC immunoreactivity was predominantly intramuscular and vascular, and appeared more pronounced in the amyloidosis patient. We then probed the potential colocalization between ASC and SAA using antibodies against ASC, SAA, and PAI-1 (as a vessel marker), as well as the amyloidotropic dye Thioflavin S. Two specific antibodies against ASC (polyclonal AL177 and monoclonal MAB/MY6745 anti-ASC, Appendix Fig. S1) and two antibodies against SAA (MAS41676 and DM003) were employed. We focused on areas in the proximity of vessels of an AA amyloidosis patient (Fig. 1C–E) and a control patient (Fig. 1F,G). The integrated density, defined as the sum of the pixel values in immunofluorescence images, of ASC was slightly increased for the patient with amyloidosis. Both antibodies targeting SAA displayed a signal increase (Fig. 1H). The thioflavin S signal colocalized with SAA in the patient with AA amyloidosis, and to a lesser degree with ASC (Appendix Fig. S2A).

We then examined regions of high SAA intensity, presumptively corresponding to amyloid aggregates, by stimulated emission depletion (STED) microscopy (Fig. 1I–L). The resulting 3D model (Fig. 1J,L) visualized the proximity of SAA and ASC within these aggregates. ASC was localized primarily in the periphery of amyloid aggregates, whereas SAA resided mostly in the amyloid cores. This differs from findings reported for ASC-Aβ colocalization, where ASC was found at the core of the plaque (Venegas et al, 2017). A graphical representation of the geolocation of the center of mass of pixels in space indicated a high distributional overlap between ASC and SAA (Appendix Fig. S2B,S2C). This provides evidence for the colocalization of ASC and SAA in human cardiac tissue affected by inflammation-induced amyloidosis.

### ASC specks interact with SAA through the pyrin domain to promote AA amyloid fibrillation

The colocalization between ASC and SAA raises the question whether ASC may promote AA amyloid formation. We therefore performed in vitro SAA aggregation assays. First, murine recombinant SAA protein (mSAA1) and Thioflavin T (ThT) were incubated in the presence of ASC specks or of bovine serum albumin (BSA) (Fig. 2A, upper panel). As we observed an accelerated amplification of mSAA1 in the presence of ASC but not of BSA, we performed a dose-response experiment with 0–500k ASC specks (Fig. 2A, lower panel). We observed that a higher number of ASC specks resulted in a left shift of the aggregation curve. In contrast, 500k ASC specks without mSAA did not induce any increase in ThT fluorescence (Fig. 2A, upper panel). To quantify the time at which the aggregation of murine SAA reached 50% of its plateau value (termed $t_{1/2}$), we performed global fitting using AmyloFit (Meisl et al, 2016) on each of the replicates individually. We found that $t_{1/2}$ was dose-dependently reduced by ASC specks (Appendix Table S1), with higher doses leading to a significant reduction in $t_{1/2}$ (adjusted $P$: 0.013–0.048, Wilcoxon rank-sum test with Holms correction for multiple comparisons, Fig. 2B). Hence the presence of ASC specks, but not of BSA, accelerates SAA fibril formation. To confirm direct interaction and to map the protein domains of interaction between ASC and SAA, we employed the

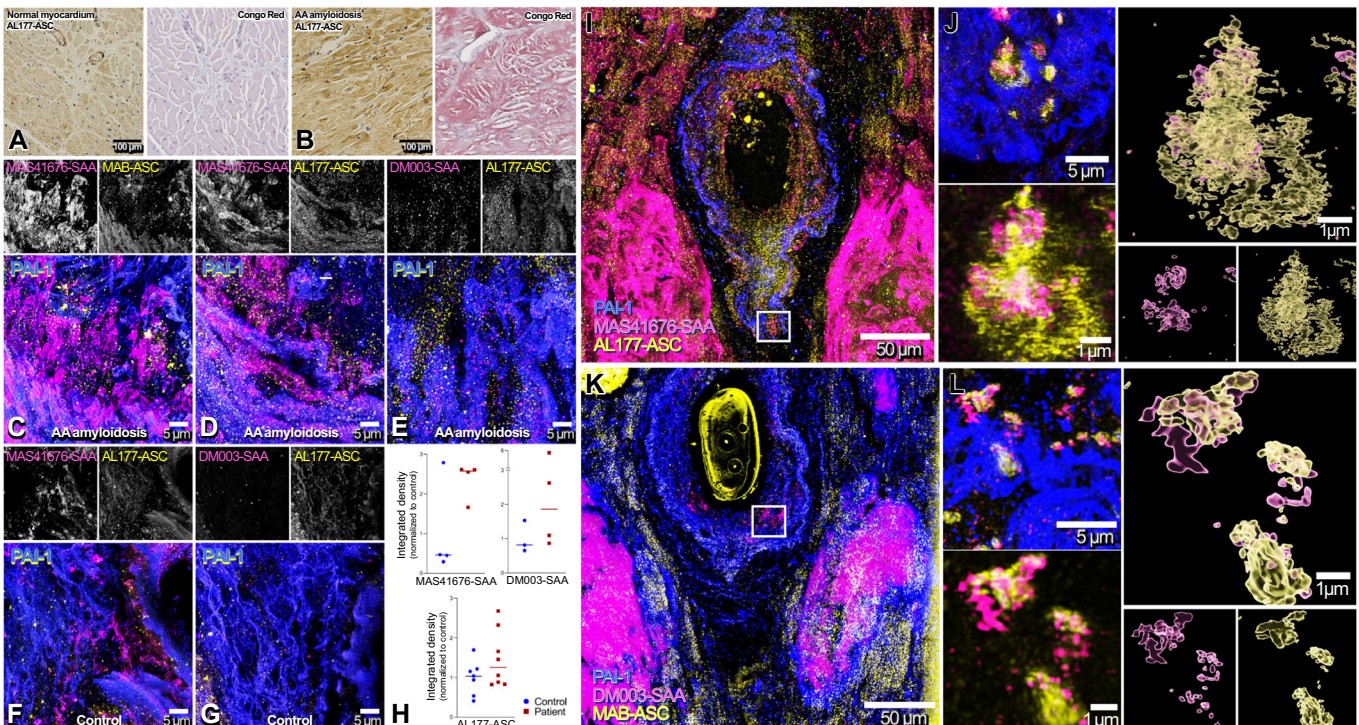

**Figure 1. Co-aggregation of ASC and SAA in human inflammation-induced amyloid A amyloidosis.**

(A, B) ASC and Congo-red-stained myocard from patients without (A) and with AA amyloidosis (B; 68 and 82-year old males, respectively). (C–E) Individual and merged confocal images of cardiac blood vessels (labeled with PAI1) from post-mortem tissue of a patient with AA amyloidosis, presenting high level of SAA (labeled with MAS41676 and DM003) and ASC (labeled with AL177 and MAB/MY6745). (F, G) Individual and merged confocal images of cardiac blood vessels (labeled with PAI1) from post-mortem tissue, presenting low level of SAA (labeled with MAS41676 and DM003) and ASC (labeled with AL177). (H) Graphical representation of the quantification of 100X confocal images for SAA and ASC. Each point represents one field of view, bars represent median. Integrated density was normalized by the mean value control. Data from one patient in each group are shown. (I) Confocal images from post-mortem tissue of a patient with AA amyloidosis showing accumulation of both SAA (MAS41676, magenta) and ASC (AL177, yellow) within the blood vessel border (labeled with PAI1) and in adjacent muscle cells. (J) STED imaging of co-aggregation SAA (magenta) and ASC (yellow). Imaris 3D model shows the intricacy of SAA and ASC within these aggregates. (K) Confocal images from post-mortem tissue of patient with AA amyloidosis showing accumulation of both SAA (DM003, magenta) and ASC (MAB-ASC (MY6745), yellow) within the blood vessel border (labeled with PAI1) and in adjacent muscle cells. (L) STED imaging of co-aggregation of SAA (magenta) and ASC (yellow). Imaris 3D model shows the intricacy of SAA and ASC within these aggregates. Source data are available online for this figure.

limited proteolysis-coupled mass spectrometry (LiP-MS) technology (Cappelletti et al, 2021; Feng et al, 2014; Pepelnjak et al, 2020; Piazza et al, 2018; Schopper et al, 2017; Schurch et al, 2022) which detects changes in peptide cleavage exerted by limited proteinase K (PK)-based proteolysis by mass spectroscopy (MS) (Holfeld et al, 2023). If an interaction between proteins exists, certain epitopes become inaccessible to PK and the peptides detected by MS will display an altered profile (Fig. 2C). First, we compared the $\log_2$ fold change of peptide representation of PK and trypsin-digested recombinant full-length ASC in the presence or absence of recombinant human SAA1 (Fig. 2D), with $|\log_2 \text{(fold change)}| > 1$ and $-\log_{10}$ (FDR-adjusted $P$ value) <0.05 as cutoffs for significance. Five significant changes in the ASC protein were identified (Fig. 2D, top, red color), all of which mapped to the pyrin domain of ASC and to part of the linker between pyrin domain (PYD) and caspase recruitment domain (CARD) (Fig. 2D, bottom), suggesting that this is the site where the protein-protein interaction with SAA occurs. The utilization of recombinant SAA1 fibrils instead of the monomeric form closely recapitulated the phenotype (Fig. 2E, top), but the linker site seemed to be more involved in the interaction (Fig. 2E, bottom). To validate the interaction between

ASC and SAA1 also in a complex cellular background, we then incubated lysate of ASC-speck-producing ExpiHEK cells with recombinant human SAA1 and performed the LiP-MS workflow on control lysate and on lysate incubated with SAA1. Among all the interactors of SAA1 (419 LiP-MS hits) in ExpiHEK cells (Fig. 2F, blue points) we detected two peptides (red points) from the same region of ASC sequence. Although the altered peptides were slightly different from those of recombinant ASC, the PYD was again structurally changed, suggesting that it represents the site of interaction with SAA1 (Fig. 2F). The PYD is also the site of interaction with Aβ (Venegas et al, 2017), whereas the CARD domain is known to interact with the CARD of Caspase-1 (Fernandes-Alnemri et al, 2007; Srinivasula et al, 2002).

## ASC does not modulate the induction of SAA by AgNO₃ or AEF

We injected AgNO₃ and amyloid-enhancing factor (AEF), consisting of preformed AA fibrils, into 12 male and 10 female wild-type (*Pycard*[+/+]), and into 14 male and 10 female B6.129-*Pycard*[tm1Vmd/tm1Vmd] (*Pycard*[−/−]) littermate mice (Fig. 3A). Mice subjected to

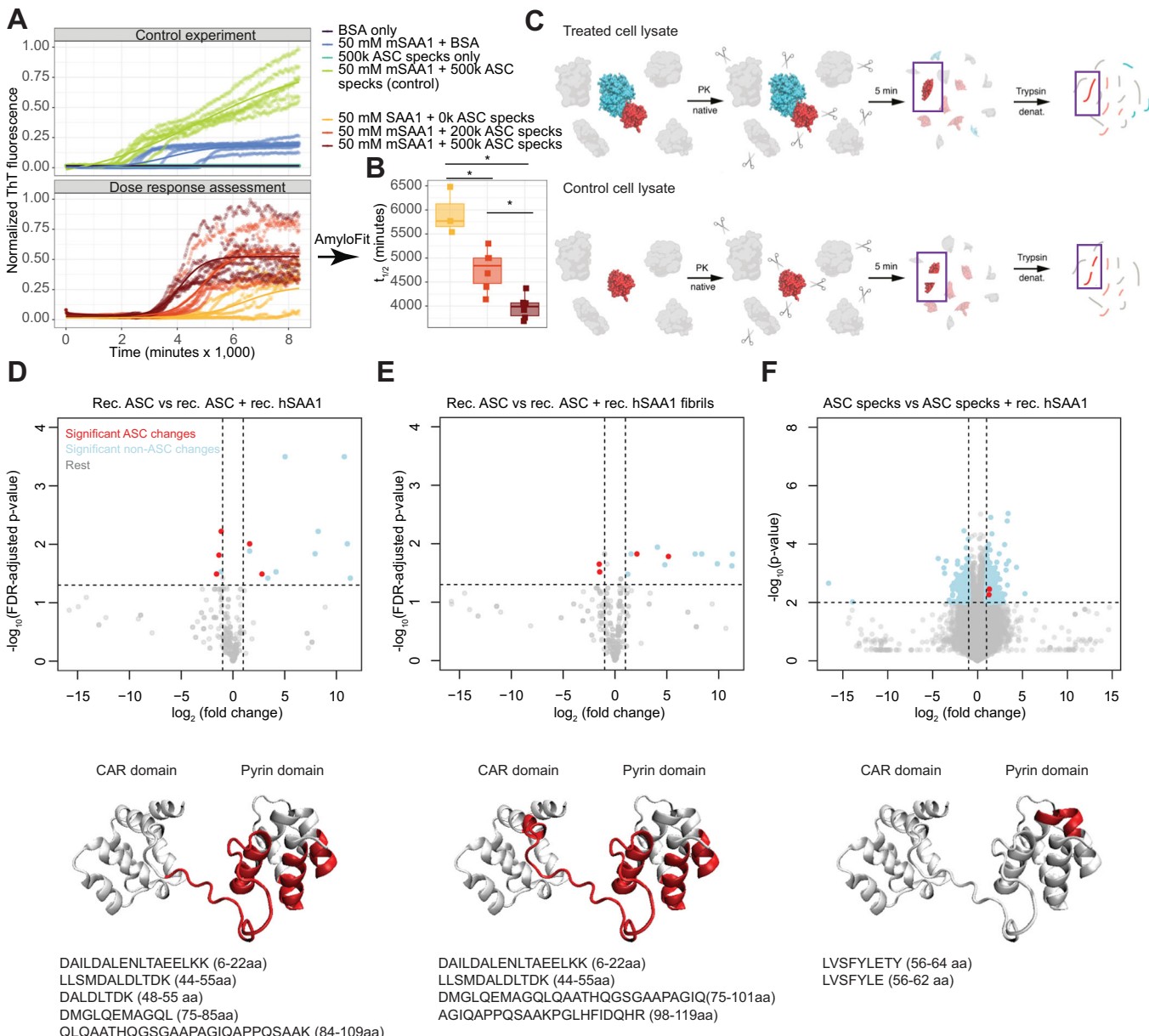

**Figure 2. ASC contacts specific SAA epitopes and accelerates SAA fibrillation.**

(A) In vitro SAA aggregation in the presence or absence of ASC specks over 150 h. Each curve represents a technical replicate. A logistic regression was performed on replicates of all conditions and the fitted curve is shown. (B) Latency to half-maximal fluorescence signal ($t_{1/2}$) for 0, 200k and 500k ASC specks. Standard boxplots: median and quartiles (Wilcoxon rank-sum test with Holm correction for multiple comparisons). Each dot represents one measurement. *$P < 0.05$ ($P$ values: 0.048; 0.048; 0.013). (C) Overview of LiP-MS. (D–F) Volcano plots (top) and ASC 3D structures (bottom) showing the LiP-MS hits and affected ASC epitopes (red) and respective amino acid sequences after mixing monomeric recombinant human ASC with monomeric recombinant human SAA1 (D), monomeric recombinant human ASC with recombinant human SAA1 fibrils (E), or human ASC specks with monomeric recombinant human SAA1 (F). Ordinate; FDR-adjusted (D, E) and unadjusted $P$ values (F); unpaired two-tailed $t$ test. (D–F) The colors denote the following: red dots—significant changes in the ASC protein. Light blue dots—significant changes other than in the ASC proteins originating from co-purified proteins. Gray dots—nonsignificant changes. Source data are available online for this figure.

AgNO$_3$ and AEF injection were denominated "AA$^+$ mice". For control, mice were injected with either PBS, AEF, or AgNO$_3$ only. These mice were collectively denominated "AA$^-$ mice".

ASC might increase AA amyloid deposition by favoring its aggregation, or by enhancing SAA induction by inflammatory stimuli. Al$^{3+}$-containing adjuvants activate the NLRP3 inflammasome, but it is not known whether AgNO$_3$ stimulates ASC-dependent inflammasomes

of innate immune cells (Eisenbarth et al, 2008), which would be a confounder in the current study. Thus, we assessed SAA levels of AgNO$_3$-only or AEF-only injected (i.e., AA$^-$) *Pycard$^{+/+}$* and *Pycard$^{-/-}$* mice by enzyme-linked immunosorbent assay (ELISA) and compared their SAA serum concentrations at baseline and up to 96 h after injection. Serum concentrations of SAA and other acute-phase proteins peaked at ~24 h after the pro-inflammatory stimulus (Fig. 3B). There

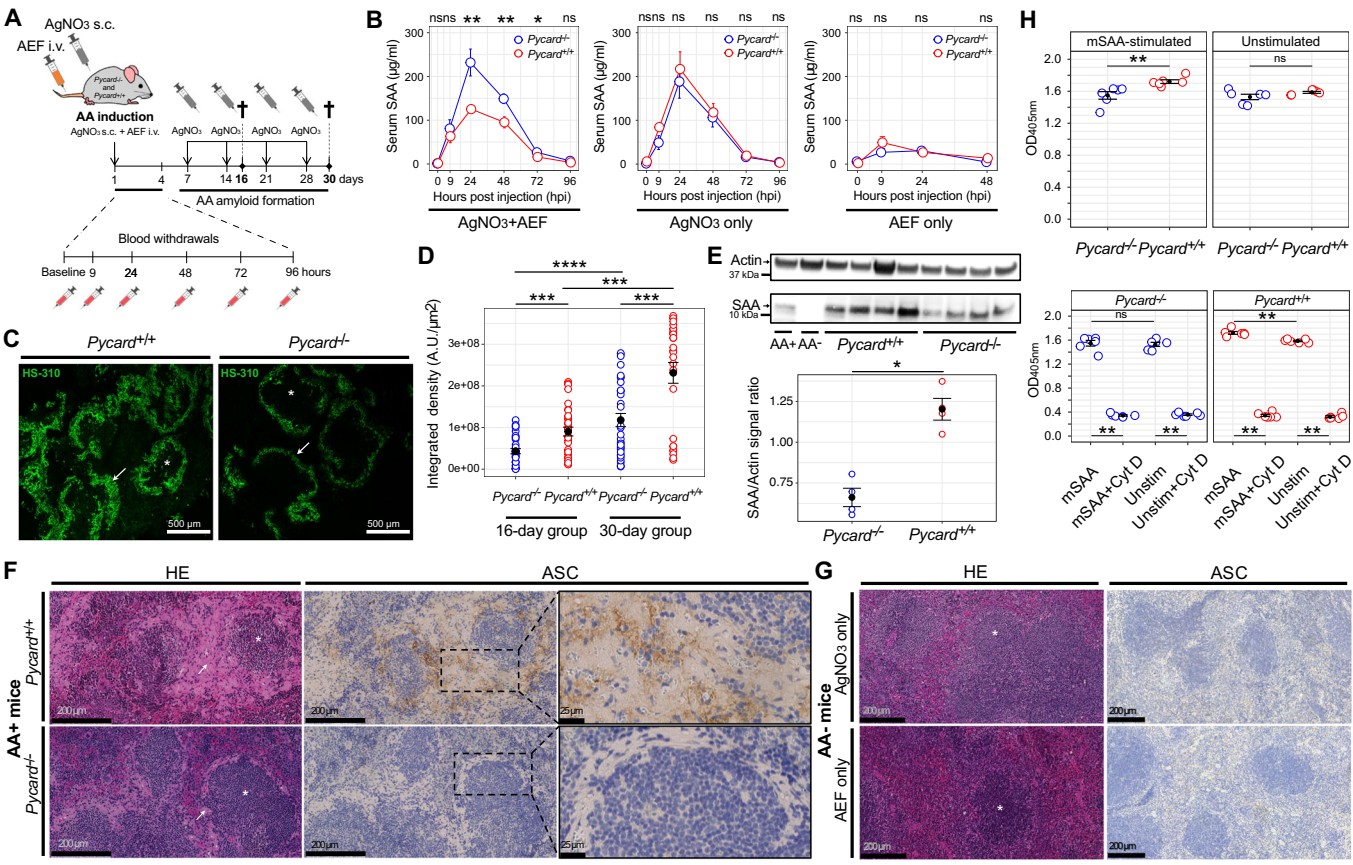

**Figure 3. Decreased splenic amyloid load in the absence of ASC.**

(A) Experimental design. Experimental groups consisted of total $n = 46$ AA$^+$ ($n = 24$ Pycard$^{-/-}$/$n = 22$ Pycard$^{+/+}$) and $n = 27$ AA$^-$ ($n = 14$ Pycard$^{-/-}$/$n = 13$ Pycard$^{+/+}$) animals ($P$ values: 0.003; 0.008; 0.03). (B) SAA serum concentrations (mean ± SEM; triplicates; Wilcoxon rank-sum test) in Pycard$^{+/+}$ and Pycard$^{-/-}$ mice after injection of AgNO$_3$, AEF, or AgNO$_3$ + AEF (AA amyloid induction). (C) AA$^+$ (Pycard$^{+/+}$ and Pycard$^{-/-}$; P30) spleens stained with LCP (green). Arrows: red pulp invasion of amyloid. Asterisks: follicles. (D) Quantification of splenic LCP signal. Dots: individual mice. Scatterplots: mean ± SEM (error bars); Wilcoxon rank-sum test with Holms correction for multiple comparisons ($P$ values: 0.002; 0.001; 0.0004; 0.002). (E) Top: Western blot of monomeric SAA in spleen homogenates of Pycard$^{+/+}$ and Pycard$^{-/-}$ mice. To ascertain SAA antibody functionality, mouse spleen homogenate from independently obtained and Congo red-confirmed AA$^+$ tissue served as positive, whereas non-induced (AA$^-$) spleen tissue served as negative technical controls. Bottom: Quantification of Western blot data, from four mice per group. mean ± SEM; Kruskal–Wallis test ($P$ value: 0.029). (F) Spleen sections stained with HE and ASC IHC from Pycard$^{+/+}$ (wt) and Pycard$^{-/-}$ AA$^+$ mice. Arrows: amorphous amyloid structures. Asterisks: splenic follicles. ASC with brown stain in IHC. Boxes are magnified in the right panels. (G) Spleen sections stained with HE and ASC from Pycard$^{+/+}$ AA$^-$ mice. (H) Phagocytic activity (OD) from unstimulated and mSAA-stimulated Pycard$^{+/+}$ and Pycard$^{-/-}$ BMDMs specimens ($P$ values; top: 0.002; bottom: 0.002). Each assay was carried out in duplicates. Cyt D: 10 μM Cytochalasin D. Mean ± SEM. Wilcoxon rank-sum test with Holms correction for multiple comparisons. Dots: individual bone marrow samples. *$P < 0.05$, **$P < 0.01$, ***$P < 0.001$, ****$P < 0.0001$, ns: not significant. Source data are available online for this figure.

was no significant difference in SAA serum concentration between Pycard$^{+/+}$ and Pycard$^{-/-}$ mice treated with only AgNO$_3$ or with only AEF at peak and at any of the time points investigated (Wilcoxon rank-sum test, Fig. 3B). Hence, the genetic ablation of Pycard does not affect AgNO$_3$-induced SAA levels, suggesting that ASC-dependent inflammasomes do not significantly contribute to AgNO$_3$ sensing. Moreover, AEF alone was insufficient to induce serum SAA levels in an ASC-dependent manner. We deduce that the anti-amyloidogenic properties of ASC ablation is not due to a modulation of SAA induction.

## AgNO$_3$-induced acute-phase responses in Pycard$^{-/-}$ and Pycard$^{+/+}$ mice

Having established the absence of a bias induced by AgNO$_3$ or with only AEF, we assessed SAA serum levels of Pycard$^{+/+}$ and Pycard$^{-/-}$ mice at baseline and 9, 24, 48, 72-, and 96-h post injection of

AgNO$_3$ and AEF (i.e., AA$^+$) (Fig. 3A). There was a significant difference in SAA levels at 24 h post injection (hpi) at the peak time of serum acute-phase protein levels (Gabay and Kushner, 1999; Kushner, 1982; Sack, 2018), between Pycard$^{+/+}$ and Pycard$^{-/-}$ AA$^+$ mice ($P = 0.004$, Wilcoxon rank-sum test, Fig. 3B). At 0–9 hpi (hours post injection), there was no difference between mean SAA serum concentrations in Pycard$^{+/+}$ and Pycard$^{-/-}$ mice. However, at 24 hpi SAA serum levels of AgNO$_3$-treated Pycard$^{+/+}$ mice were significantly higher than in Pycard$^{+/+}$ mice treated with AgNO$_3$ + AEF ($P = 0.03$, Wilcoxon rank-sum test). This suggests a "sink effect" by which SAA is recruited by AEF and/or by nascent amyloid, and therefore decreases in serum. Crucially, this reduction in SAA serum concentration did not occur in Pycard$^{-/-}$ AA$^+$ mice. The mean SAA serum concentrations of Pycard$^{-/-}$ AA$^+$ mice were marginally higher than those of AgNO$_3$-only injected Pycard$^{-/-}$ groups at 9, 24, 48, and 72 hpi, possibly because the ELISA detected

some inoculated AA fibrils that may still have been present in the bloodstream. At 48 hpi, SAA serum concentrations decreased in all experimental groups. However, at this time point the difference between $Pycard^{+/+}$ and $Pycard^{-/-}$ AA$^+$ mice was still significant ($P = 0.009$, Wilcoxon rank-sum test, Fig. 3B). Depending on the genotype, the difference between SAA serum concentrations in $Pycard^{+/+}$ and $Pycard^{-/-}$ AA$^+$ mice persisted up to 72 h after injections ($P = 0.03$, Fig. 3B). Finally, 96 h after injection there was no difference in SAA serum concentration. $Pycard^{-/-}$ AA$^+$ mice were found to have the highest SAA serum concentration among all experimental groups (Appendix Table S2). We conclude that ASC facilitates the recruitment and deposition of SAA in murine inflammation-associated amyloidosis.

## Decreased splenic amyloid deposition in the absence of ASC-dependent inflammasomes

We assessed the presence of amyloid in spleens of mice with experimental AA amyloidosis (Fig. 3A) by staining histological sections of paraffin-embedded tissue with Congo Red (CR) (Appendix Fig. S3) and with the luminescent conjugated poly-thiophene (LCP) HS-310 (Fig. 3C; Appendix Fig. S4). We sacrificed experimental animals at day 16 or 30, 2 days after the last AgNO$_3$ injection. CR-stained amyloid showed the characteristic red appearance in bright-field microscopy and apple-green birefringence under polarized light, similarly to human myocardial tissue (Fig. 1). The red spleen pulp of $Pycard^{+/+}$ AA$^+$ mice exhibited more pronounced amyloid invasion than that of $Pycard^{-/-}$ AA$^+$ mice (white arrows), and no CR and LCP staining was detected in AA$^-$ mice (Appendix Figs. S3 and S4). The quantification of HS-310 fluorescence intensity confirmed a significant difference in the amyloid load of $Pycard^{+/+}$ and $Pycard^{-/-}$ AA$^+$ mice at day 16 and 30 (adjusted $P = 0.002$ and $0.001$, respectively; Wilcoxon rank-sum test with Holms correction for multiple comparisons) indicating that AA deposition is ASC-dependent (Fig. 3D; Appendix Table S3). The median difference increased over time from 2.3-fold (interquartile range (IQR): 0.8-2.7) at day 16 to 3.4-fold (IQR: 1.5–6.9) at day 30.

Using western blot (WB), we assessed the presence of the AA amyloid precursor protein SAA in the spleen homogenate of the four $Pycard^{+/+}$ and $Pycard^{-/-}$ AA$^+$ mice with the highest splenic amyloid load of the 30-days group (Fig. 3E). We observed a reduction of total splenic SAA in $Pycard^{-/-}$ AA$^+$ mice. The median reduction was 1.9-fold (IQR: 1.7–2.1). Like human AA aggregates, we found ASC immunoreactivity adjacent to amyloid deposits (Fig. 3F). No amyloid or ASC signals were detected in AA$^-$ mice of similar age injected with either PBS, AgNO$_3$ or AEF (Fig. 3G; Appendix Fig. S5). ASC immunoreactivity was mostly detected in the perifollicular region of $Pycard^{+/+}$ AA$^+$ spleens (Fig. 3F). These data indicate that the AA amyloid load is strongly modulated by $Pycard$.

## No specific transcriptional signature in macrophages upon amyloidosis induction

Prolonged elevation of serum SAA is required to trigger AA amyloidosis (Simons et al, 2013). AA aggregates then progressively disrupt tissue integrity and ultimately impair the physiological function of affected organs. Furthermore, patients with systemic AA amyloidosis may exhibit altered red blood cell and platelet

volumes (Bakan et al, 2019; Erdem et al, 2014), which in turn can activate inflammasomes and boost the inflammasome capacity of macrophages, neutrophils and monocytes (Rolfes et al, 2020). We therefore assessed the cellular composition of the spleen and of peripheral blood at baseline and after induction of experimental AA amyloidosis in $Pycard^{+/+}$ and $Pycard^{-/-}$ mice.

Using flow cytometry, we counted splenic B cells, T cells, dendritic cells, neutrophils as well as M1- and M2-like macrophages. Mice that had only been treated with AgNO$_3$ and AA$^+$ mice showed increased splenic macrophage infiltration compared to baseline. Macrophage infiltration was neither dominated by M1-nor by M2-like macrophages (Appendix Fig. S5). We assessed the transcriptional state of fluorescence-activated cell sorted (FACS) splenic macrophages from $Pycard^{+/+}$ and $Pycard^{-/-}$ AA$^+$ mice by RNA sequencing (Appendix Fig. S6A, S6B). Significant ($P < 0.05$; FDR $< 0.01$) transcriptional changes were only found in the $Pycard$ gene, as expected, as well as in $Gdpd3$ (Appendix Fig. S6C,S6D), a glycerophosphodiester phosphodiesterase of unknown relevance in the context of SAA. The absence of a distinct macrophage signature in inflammation-associated amyloidosis suggests that other inflammation-related proteins, such as cytokines, may not play an important role in AA formation and deposition.

## Reduced phagocytic activity of SAA-activated Pycard$^{-/-}$ bone marrow-derived macrophages

Macrophages and monocytes play a central role in AA amyloidosis. They co-localize with AA amyloid in the spleen of AA$^+$ mice (Sponarova et al, 2013) and can transfer AA amyloidosis in vivo (Sponarova et al, 2008). Conversely, phagocyte depletion delays or inhibits AA amyloid accumulation (Kennel et al, 2014; Lundmark et al, 2013). Moreover, AA amyloid undergoes Fcγ-receptor-mediated phagocytosis by macrophages, which is initiated by host-specific antibodies that target the AA protein (Nyström and Westermark, 2012). Since SAA activates macrophages (Chen et al, 2020; Gaiser et al, 2021; Niemi et al, 2011), and $Pycard^{+/-}$ astrocytes of APP/PS1 transgenic mice overexpressing mutant amyloid β precursor protein and presenilin-1 (a mouse model of Alzheimer's disease) show increased Aβ phagocytosis (Couturier et al, 2016), we investigated whether the presence or absence of ASC, in the context of SAA induction, influences the phagocytic activity of murine bone marrow-derived macrophages (BMDMs) in vitro (Appendix Fig. S7A,S7B). We exposed SAA-activated and non-SAA-activated murine BMDMs to an in vitro phagocytosis assay. There was no significant difference in phagocytic activity between unstimulated $Pycard^{+/+}$ and $Pycard^{-/-}$ BMDMs ($P = 0.4$, Kruskal–Wallis test) (Fig. 3H). However, activity was higher in SAA-stimulated than in unstimulated $Pycard^{+/+}$ BMDMs ($P = 0.002$) but not in SAA-stimulated compared with unstimulated $Pycard^{-/-}$ BMDMs ($P = 0.589$), suggesting that SAA is partially ASC-dependent in triggering phagocytosis. Furthermore, SAA-stimulated $Pycard^{+/+}$ BMDMs showed higher phagocytic activity than $Pycard^{-/-}$ BMDMs ($P = 0.002$), underlining a functional implication of the ASC protein (Appendix Table S4).

We considered that the ablation of $Pycard$ may cause pathologies of cellular compartments that do not require a direct ASC–SAA interaction. We assessed lymphocyte, monocyte, granulocyte, red blood cell and platelet counts in peripheral blood of AA$^+$ and AA$^-$ experimental animals (Appendix Figs. S8 and S9). Of special interest

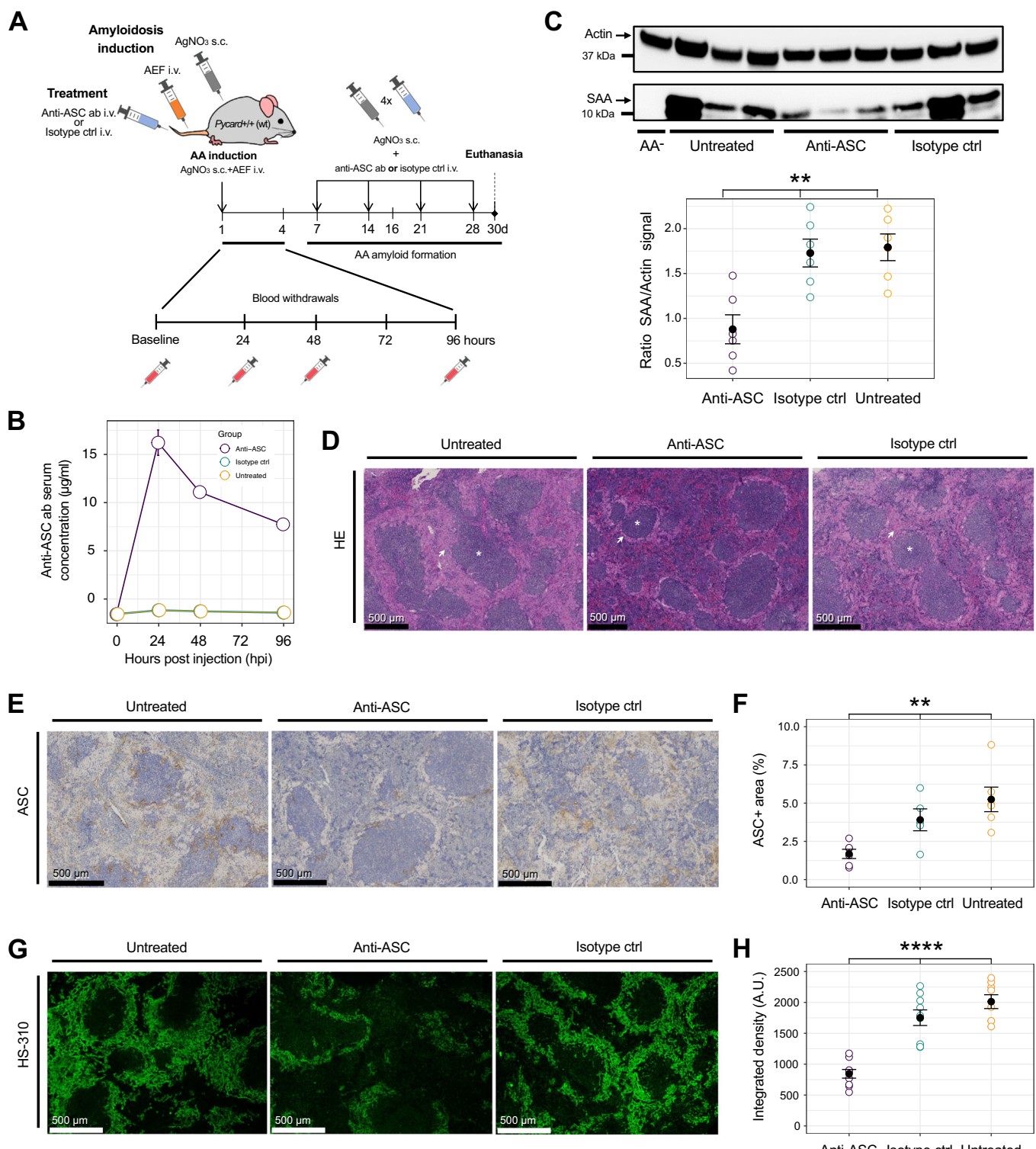

were myeloid cells as well as platelets. The former form the cellular substrate of *Pycard*-dependent inflammasomes (Pandey et al, 2021), whereas the latter were shown to be key for enhancing inflammasome activation and capacity of innate immune cells upon PRR stimulation (Rolfes et al, 2020). Platelet count at baseline was higher in *Pycard*$^{-/-}$ than in *Pycard*$^{+/+}$ mice ($P = 0.017$). The same was observed for *Pycard*$^{-/-}$ AA+ mice from the 30-days group ($P = 0.036$, Appendix Fig. S9). The reduction of SAA-induced phagocytosis in *Pycard*$^{-/-}$ BMDMs (compared to *Pycard*$^{+/+}$ BMDMs) supports the notion that ASC is responsible for the modulation of AA amyloid deposition.

◄ **Figure 4. Anti-ASC^PYD immunotherapy reduces peripheral amyloid load.**

(A) Scheme of anti-ASC immunotherapy. Total $n = 15$ experimental AA$^+$ *Pycard*$^{+/+}$ (wt) animals ($n = 5$ per treatment group). (B) Anti-ASC antibody single-injection pharmacokinetics. $N = 3$ per treatment group, technical duplicates; mean ± SEM. (C) Western blot of SAA in spleen homogenate of *Pycard*$^{+/+}$ AA$^+$ mice treated with anti-ASC or isotype antibody, or untreated. Absence of amyloidosis induction as control (AA$^-$). Actin and spleen homogenate from a C57BL/6 wt mouse served as negative control. Scatterplots show data from three mice/group and two independently performed immunoblots resulting in 6 samples and data points (technical duplicates; mean ± SEM, Kruskal–Wallis test, P value: 0.005). (D) HE-stained spleen sections of *Pycard*$^{+/+}$ AA$^+$ mice. Arrows: amyloid. Asterisks: white pulp. (E) Spleen sectiond stained with ASC antibodies (brown). (F) Quantification of ASC-positive areas in spleen sections. Two fields of view, each representing a different section for each mouse (3 mice/group) were quantified. Scatterplots: mean ± SEM. Statistics: Kruskal–Wallis test, P value: 0.007. (G) Spleen sections stained with LCP. (H) Integrated density expressed as arbitrary units/μm² in LCP-stained spleen sections. Each dot represents a quantified field of view. Three field of views for each animal were analyzed (nine fields of view from three mice per group). Scatterplots: mean ± SEM; Kruskal–Wallis test, P value: 0.0001. **$P \le 0.01$, ****$P \le 0.0001$. Source data are available online for this figure.

## Anti-ASC immunotherapy diminishes inflammation-induced amyloid deposition

Therapeutic antibodies can interfere powerfully with amyloid formation (Schenk, 2002). We therefore asked whether treatment with anti-ASC antibodies can reduce AA amyloidogenesis in mice (Fig. 4A). We generated a rabbit monoclonal anti-murine ASC antibody targeting the pyrin domain (PYD) and replaced its rabbit Fcγ domain with that of a mouse IgG$_{2a}$ domain to avoid xenogeneic anti-drug responses in recipients and to improve its effector functions in vivo. The engineered antibody (denoted MY6745) showed high affinity binding to murine ASC$^{PYD}$ (Appendix Fig. S10A). Purified, endotoxin-free mouse monoclonal anti-murine ASC and mouse monoclonal isotype control antibodies were diluted in PBS to 2 mg/ml and intravenously injected (5 mg/kg body weight) at an interval of 7 days. Anti-ASC antibody serum levels were determined at various time points up to 96 hpi after the first injection and were found to peak at 24 hpi. At 96 hpi, anti-ASC antibody serum levels declined to approximately half-maximal serum concentration levels, reflecting the expected half-life of therapeutic IgG (Fig. 4B). Western blotting revealed that treatment of *Pycard*$^{+/+}$ AA$^+$ mice with antibody MY6745, but not with a non-specific monoclonal isotype-control antibody, reduced total splenic SAA 2.3-fold (IQR: 1.7–3.0, $P = 0.005$, Kruskal–Wallis test, Fig. 4C). Following qualitative amyloid assessment on HE-stained sections (Fig. 4D) we performed ASC immunohistochemistry. The anti-ASC antibody treatment significantly diminished the ASC immunohistochemical signal in spleens of *Pycard*$^{+/+}$ AA$^+$ mice (Fig. 4E), indicating effective target engagement and modulation. The groups significantly differed ($P = 9.9 \times 10^{-5}$, Kruskal–Wallis test, Fig. 4F) and the median reduction was 2.7-fold (IQR: 2.4–4.8) compared to the untreated group (with 1.3-fold (IQR: 1.0–1.4) reduction for of the isotype-treated group). Following a quantitative assessment of mature extracellular amyloid, LCP-stained (Fig. 4G) spleen sections showed a conspicuous reduction in amyloid deposition of anti-ASC treated AA$^+$ mice ($P = 0.007$, Kruskal–Wallis test, Fig. 4H). Importantly, anti-ASC antibody treatment had a strong effect, with a 2.5-fold (IQR: 2.4–3.3) reduction compared to the untreated condition, whereas the isotype-treated group only displayed a 1.2-fold (IQR: 1.1–1.7) reduction in amyloid deposition. Furthermore, AA$^+$ mice treated with anti-ASC antibodies experienced a trend toward less severe loss of body weight than non-treated isotype-antibody-injected AA$^+$ mice, although the differences were not statistically significant (Kruskal–Wallis test, Appendix Fig. S10B). We conclude that anti-ASC treatment reduces AA amyloid deposition similarly to the ablation of *Pycard*.

## A large-scale investigation of anti-ASC autoantibodies in unselected hospital patients advocates stringent immune tolerance against ASC

Autoantibodies against aggregation-prone proteins can be beneficial against pathological protein aggregate but can also precipitate disease (Senatore et al, 2020; Sevigny et al, 2016; Sonati et al, 2013). Having established that ASC controls the extent of amyloid deposition in inflammation-associated amyloidosis in mice and that ASC interacts with SAA via its PYD, we investigated whether natural anti-ASC antibodies exist in human antibody repertoires. Such autoantibodies could modulate systemic amyloidosis or conceivably other forms of protein aggregation. First, we established a high-throughput micro-ELISA using an automated robotic platform (Emmenegger et al, 2023a), using an ASC protein containing a C-terminal his-tag (Venegas et al, 2017) at a concentration of 1 μg/ml at a well volume of 3 μl for coating. We then interrogated an unselected cohort consisting of 23,450 plasma samples from 19,334 patients admitted to various clinics at the University Hospital Zurich (Fig. 5A) for the presence of autoantibodies against monomeric ASC protein. The median age of the patients was 55 (interquartile range: 39–68) years and the female:male ratio was 47.3:52.7 (Fig. 5B). The largest fraction of samples originated from the department of cardiology, followed by dermatology, rheumatology, and hematology. We first compared the reactivity titers, i.e., the $-\log_{10}(EC_{50})$ or, in short, p(EC$_{50}$) values, of all patients with that of 119 patients with a history of amyloidosis based on ICD-10 code E85 ("Amyloidosis—excluding Alzheimer's disease"), including five individuals with AA amyloidosis, and with 21 patients with ICD-10 codes G30 or F00 ("Alzheimer's disease" and "Dementia with Alzheimer's Disease"). None of the profiles protruded significantly from the collective (Fig. 5C, Kruskal–Wallis test with post-hoc Wilcoxon rank-sum test). We then conducted an exploratory logistic regression analysis using a Bayesian LASSO (Emmenegger et al, 2023a; Emmenegger et al, 2023b; Lamparter et al, 2022) on the entire dataset to identify whether, and which, ICD-10 codes are associated with seropositivity. At high specificity (p(EC$_{50}$) threshold value 2), the highest odds ratios were seen with ICD-10 codes E87 ("Other disorders of fluid, electrolyte and acid-base balance"), K58 ("irritable bowel syndrome"), and S69 ("Other and unspecified injuries of wrist and hand"). However, their odds ratios were 1.00 with 95% credible intervals of 0.78–1.61, indicating no effect. We additionally conducted a sensitivity analysis where we lowered the p(EC$_{50}$) cutoffs to 1.7 and 1.5, respectively. While lower cutoffs necessarily decrease the specificity of this analysis, we aimed to explore

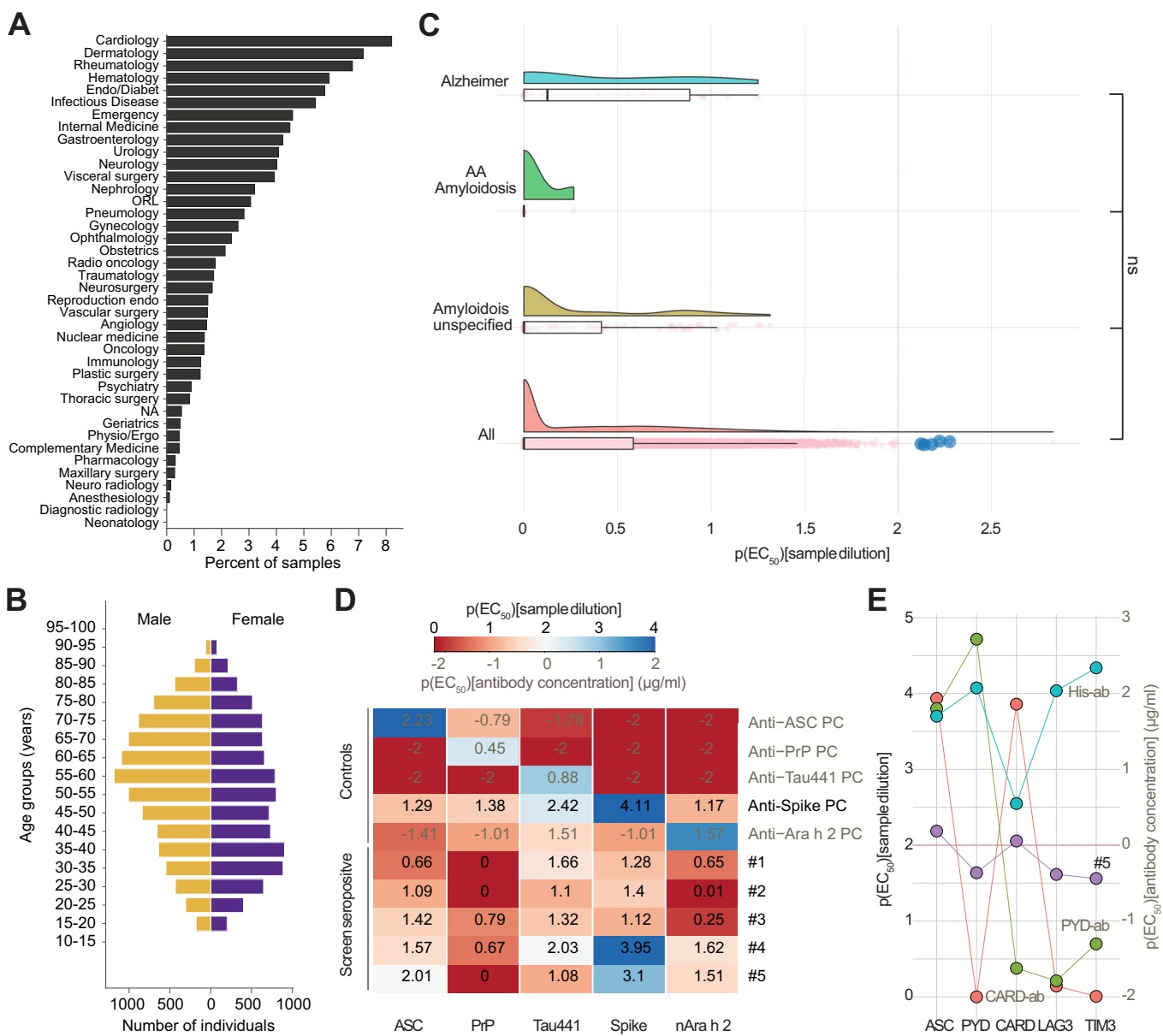

**Figure 5. Population-wide interrogation of autoantibodies against ASC in a large unselected hospital cohort advocates stringent immune tolerance against ASC.**

(A) Sample provenance by various hospital units. The contributions are depicted as percentage. (B) Age distribution of subjects. (C) Raincloud plots which includes standard boxplots (median and quartiles) displaying jittered $p(EC_{50})$ values for all patients screened (All), for the fraction of patients characterized by ICD-10 code E85 (Amyloidosis), AA amyloidosis, or Alzheimer's disease. The dashed red line at $p(EC_{50}) = 2$: reactivity cutoff. Blue dots represent the hits, characterized by $p(EC_{50}) \geq 2$ and a mean squared residual error < 20% of the actual $p(EC_{50})$. None of the groups were significantly different (Kruskal–Wallis test, $P$ value = 0.079, $\alpha = 0.01$; Wilcoxon rank-sum test $P$ value after Holm correction for multiple comparisons: 0.43 (All vs. Amyloidosis), 0.47 (All vs. AA Amyloidosis), 0.47 (Amyloidosis vs. AA Amyloidosis), 0.47 (All vs. Alzheimer), 0.28 (Amyloidosis vs. Alzheimer), 0.43 (AA Amyloidosis vs. Alzheimer), $\alpha = 0.01$. (D) Heatmap showing $p(EC_{50})$ values of seropositive samples from screen and controls assayed against an array of antigens. Only the plasma of patient #5 exceeded the cutoff value and was a confirmed binder of human ASC. All samples and controls were assayed as duplicates. (E) Patient #5 (purple), along with multiple control antibodies, was assayed against the full-length ASC protein (ASC), the PYD of ASC, the CARD of ASC, and against LAG3 and TIM3. All proteins used contained a his-tag. All samples and controls were assayed as duplicates. (D, E) Two scales are given: (1) for plasma samples, the $p(EC_{50})$ of the respective plasma dilution is used. (2) For monoclonal antibodies of known concentration, the $p(EC_{50})$ of the concentrations (in µg/ml) are shown. Anti-ASC, Anti-PrP, and anti-Tau441-positive control (PC) as well as CARD-ab, PYD-ab, and His-ab are monoclonal antibodies. Source data are available online for this figure.

whether noteworthy differences would manifest more markedly. At $p(EC_{50})$ threshold value 1.7, the highest odds ratios were seen with ICD-10 codes J98 ("Other respiratory disorders"), E83 ("Disorders of mineral metabolism"), and R04 ("Hemorrhage from respiratory passages"). However, their odds ratios were 1.00 with 95% credible intervals of 0.86–1.64. Only when lowering the $p(EC_{50})$ threshold even further to values that cannot be considered biologically meaningful, some odds ratios started becoming different from 1.00. The three ICD-10 codes most associated with positivity were U99 ("Special screening examination for SARS-CoV-2"), K65

("Peritonitis"), and E84 ("Cystic fibrosis"), with odds ratios of 1.88, 1.33, and 1.31, respectively (see Appendix Fig. S11). However, these odds ratios entail large 95% credible intervals between 0.85 and 3.84. From this data exploration, we conclude that seropositivity at different cutoffs is not significantly associated with disease conditions.

By defining, based on previous studies (Senatore et al, 2020), patients with $p(EC_{50})$ values ≥ 2 with respective fitting error <20% as seropositive we identified five candidates who exceeded the threshold in the high-throughput screen (Fig. 5C, blue dots). The five seropositive candidates were then tested for specificity and reproducibility with an antigen panel. Only patient #5 confirmed reactivity against the ASC protein (Fig. 5D). The same patient did not show binding to control proteins such as the human recombinant prion protein ($PrP_{23-230}$ or PrP), full-length tau (Tau441), or the natural Ara h 2 allergen but had received multiple SARS-CoV-2 vaccinations, which explains the positivity to the SARS-CoV-2 Spike protein. We then conducted an additional experiment on patient #5, to detail the epitope and account for potentially unspecific binding to the C-terminal his-tag on the ASC protein. Patient #5 displayed $p(EC_{50})$ values ≥ 2 against the full-length ASC protein as well as against the ASC CARD but not against the ASC PYD, or against his-tagged LAG3 or TIM3 control proteins (Fig. 5E). These results suggest that this patient, admitted to the hospital because of an acute relapsing tonsillitis, had elicited a genuine immune response targeting the CARD of the ASC protein. We conclude that the humoral response against the human monomeric ASC protein can indeed occur but is exceedingly rare (<0.01%). This indicates a stringent immune tolerance of the human immune system to ASC.

# Discussion

Here we report that ASC, the central component of the NLRP3, NLRC4 and AIM2 inflammasomes (de Alba, 2019; Mariathasan et al, 2004), coaggregates with SAA in tissues of patients with inflammation-associated AA amyloidosis. We interrogated the role of ASC in inflammatory AA amyloidosis using a variety of experimental platforms including super-resolution microscopy, AA fibrillization in vitro, limited-proteolysis mass spectrometry, and a murine model of inflammation-induced AA amyloidosis. In addition, we explored the potential of immunotherapy against ASC as a pharmacological approach against inflammation-related amyloidosis. Finally, we investigated the prevalence of naturally occurring anti-human ASC-autoantibodies in a large-scale screen of an unselected human population. All the above studies indicate that ASC has a profound effect on the deposition of AA amyloid by interacting with SAA via its pyrin domain.

We used STED super-resolution microscopy to investigate the topological relationship between ASC and SAA. In contrast to amyloid β (Venegas et al, 2017), ASC co-aggregated in the periphery of AA amyloid, suggesting that ASC and SAA co-localize. But does ASC physically interact with SAA, and if yes, through which epitope? Using aggregation assays and LiP-MS, we found that the ASC-SAA interaction occurs at a few epitopes located in the pyrin domain of the ASC protein. This finding reflects the literature as the ASC pyrin domain, besides its well-known ASC-homodimerization characteristics (Fernandes-Alnemri et al, 2007; Oroz et al, 2016; Vajjhala et al,

2012), is described to be a protein interaction domain (de Alba, 2019; Venegas et al, 2017). Therefore, the pyrin domain may represent an attractive target for therapy development.

In $Pycard^{+/+}$ $AA^+$ mice, the deposition of AA amyloid appeared to be more invasive into the red pulp than in $Pycard^{-/-}$ $AA^+$ mice. This suggests that ASC inflammasome protein assemblies, driven by $ASC^{PYD}$ oligomerization and $ASC^{CARD}$ cross-linking, modulate amyloidogenesis and/or the deposition of AA amyloids in tissue, similarly to what was found for amyloid β (Venegas et al, 2017) and tau (Ising et al, 2019). It is thus conceivable that ASC serves as a generic protein aggregation-enhancing platform in various central and peripheral PMDs where the innate immune system is involved.

We probed alternative routes by which ASC could decrease the amyloid load. It is widely recognized that macrophages co-localize with amyloid deposits and that Fcγ-receptor-mediated phagocytosis plays an important role in deposition, processing, and clearing of AA amyloid (Lundmark et al, 2013; Nyström and Westermark, 2012; Richards et al, 2018; Sponarova et al, 2013; Vahdat Shariat Panahi et al, 2019). We found that $Pycard^{+/+}$ murine BMDMs showed stronger phagocytosis than $Pycard^{-/-}$ BMDMs, similarly to phagocytosis of Aβ in a murine model of AD (Couturier et al, 2016). In addition, transcriptomic assessment of splenic AA-diseased mice macrophages only revealed few differentially expressed genes between $Pycard^{+/+}$ and $Pycard^{-/-}$ animals. As expected, ASC transcripts were only identified in $Pycard^{+/+}$ mice. Moreover, increased splenic macrophage infiltration in $AgNO_3$-only ($AA^-$ mice) as well $AA^+$ mice, compared to baseline levels, further underscores the involvement of innate immune cell populations, especially myeloid cells, in the pathogenesis of AA amyloidosis. These findings suggest that the reduced splenic AA amyloid load in $Pycard^{-/-}$ mice is mainly due to the absence of ASC rather than enhanced phagocytosis of $Pycard^{-/-}$ BMDMs, even if there may be post-translational signatures or transcriptional changes that went unnoticed owing to limited sensitivity.

Another contributor to AA formation in chronic inflammation may be HDL-SAA complexes forming when entering the circulation. An in vitro ThT-fluorescence aggregation assay showed that HDL protects SAA from aggregation (Claus et al, 2017). HDL may sequester SAA in an α-helical dominant conformational state, thereby preventing fibrillogenic. Moreover, HDL-SAA complexes and monomeric SAA have been shown to be endocytosed into the lysosomal compartment of macrophages, where complex dissociation and self-assembly of SAA into AA protofibrils is initiated (Claus et al, 2017; Rocken and Kisilevsky, 1998). ASC may therefore not only interact with SAA but also with HDL–SAA complexes. This interaction may lead to SAA release from these complexes, shifting the equilibrium toward AA formation. Moreover, post-translational modifications such as phosphorylation and ubiquitination of ASC (CARD and PYD) regulate ASC assembly and inflammasome activity in a complex manner (Moretti and Blander, 2021). The sum of evidence, therefore, suggests that ASC-dependent acceleration of amyloid genesis is due to physical amyloid scaffolding and/or to enhanced macrophage-dependent spread of seeds (Gaiser et al, 2021).

Having discovered that the ASC pyrin domain physically interacts with SAA, we hypothesized that a pharmacological targeting of this domain might recapitulate the results seen in the *Pycard* knockout model, i.e., reduced amyloid deposition. Indeed, anti-$ASC^{PYD}$ antibodies were effective in reducing amyloid load. In this context, anti-$ASC^{PYD}$

interventions may impact the immune function indirectly and reduce the visceral AA amyloid load, similar to therapies combining small molecules targeting circulating serum amyloid P component (SAP) with monoclonal anti-SAP antibodies (Bodin et al, 2010; Richards et al, 2018). Anti-ASC antibodies were reported to be effective in vivo by binding intracellular ASC filaments and suppressing the immune-inflammatory response driving the development and progression of multiple sclerosis (de Rivero Vaccari et al, 2023; Desu et al, 2020). Pharmacological targeting of the ASC[PYD] adapter possesses therapeutic potential as IL-1 independent ASC inflammasome properties contribute to the pathogenesis of various proteinopathies.

Autoantibodies against endogenous proteins can modulate disparate pathologies (Goncalves et al, 2021; van der Wijst et al, 2021). Therefore, in a large-scale anti-ASC autoantibody screen of cross-departmental hospital patients, we investigated the humoral B cell response against human ASC. Such autoantibodies could neutralize ASC specks and impair AA amyloid formation. However, such humoral responses were exceedingly rare (<0.01%), indicating a high tolerance of the human immune system to ASC. Thus, natural anti-ASC autoantibodies are unlikely to play a key role in the modulation of inflammation-associated amyloidosis. Conversely, the absence of a natural immune response to ASC may increase the chance of high tolerance as well as successful immunotherapy approaches employing anti-ASC antibodies in inflammatory diseases. Moreover, the magnitude of anti-ASC titers was not associated with any specific clinical diagnoses.

In conclusion, we report a crucial role for ASC in SAA interaction and recruitment, SAA serum level modulation, SAA fibril formation acceleration, and controlling the extent of inflammation-associated amyloidosis with respect to peripheral AA amyloid deposition. It is conceivable that the effect is not only a result of the binding properties of ASC and ASC specks for β-sheet-rich proteins. Alternatively, ASC assemblies may serve as aggregation-enhancing scaffolds in various PMDs. Our findings might have therapeutic implications that advance the fields PMDs and chronic inflammatory diseases in general as ASC could be a target of disease-modifying therapies that aim to reduce amyloid deposition and pathology in various proteinopathies.

# Methods

## Histology and immunohistochemistry of human tissue

Formalin-fixed and paraffin-embedded cardiac sections (2–4 μm) were de-paraffinized with three cycles of xylene treatment followed by re-hydration with 100% EtOH, 96% EtOH, 70% EtOH, and water (each cycle 5 min), respectively. Hematoxylin and eosin (HE) as well as Congo red (CR) staining and ASC immunohistochemistry (IHC) were performed according to standard procedures used at our institute. IHC was performed with anti-ASC pAb (clone AL177, AdipoGen) at a dilution of 1:500.

For confocal and stimulated emission depletion (STED) microscopy, formalin-fixed and paraffin-embedded cardiac sections (15 μm) were de-paraffinized with xylene, followed by re-hydration with 100% EtOH, 95% EtOH, 80% EtOH, 70% EtOH and 50%EtOH. Sections were blocked with a 10% donkey serum, 5% BSA PBS 0.02% triton. PAI1 (PA1-9077, ThermoFisher), SAA (MAS41676, Invitrogen), ASC (clone AL177, AdipoGen) and ASC (MY6745, Mabylon) were used at 1:200

dilution, SAA (DM003, OriGene Technologies) was used at 1:20 dilution, all in saturating buffer, for 48 h at 4 °C. Alexa 488, 568 and 647 from Invitrogen were used at 1:200 concentration in saturating buffer, for 24 h at 4 °C. Thioflavin S staining (1%) was performed for 15 min at room temperature. Sections were treated with Sudan Black before mounting in ProLong Diamond Antifade Mountant (ThermoFisher).

Confocal and super-resolution STED images were acquired on a Leica SP8 3D, 3-color-gated STED laser scanning confocal microscope. 10x (NA 0.3, Appendix Fig. S2), 20× (immersion Oil, NA 0.75) and 100x (immersion oil, NA 1.4) images were acquired at a resolution of 2048x2048. In all, 10× images were acquired with a 600 Hz bidirectional scan yielding a XY resolution of 567.9 μm per pixel, 20× images were acquired with a 700 Hz bidirectional scan yielding a XY resolution of 126.2 μm per pixel, and 100x images with a 1000 Hz bidirectional scan yielding a XY resolution of 25.2 μm per pixel. Images were then deconvoluted using Huygens Professional software, and analyzed with image J Fiji. All presented images are max Z-projection and were adjusted for gray values identically for control and AA amyloidosis patient. STED images were visualized using Imaris 10.0.0 version.

## Culturing and immunocytochemistry of THP-1 cells

To assess the specificity of ASC antibodies (clone AL177, AdipoGen; clone MY6745, Mabylon AG), wild-type THP-1 (human monocytic leukemia cell line) and ASC-knockout THP-1 cells (Mabylon) were stained using immunocytochemistry (ICC). Additional controls included ASC/TMS1 (clone 1C3D7, Novus Biologicals), TMS1/ASC (RM1049, Abcam), and rabbit IgG isotype control (AG-35B-0013, AdipoGen) antibodies. THP-1 cells were cultured according to standard protocols. For ICC, THP-1 cells (0.5 × 10⁶/mL) were added to poly-*D*-lysine-coated glass coverslips in a 12-well plate and allowed to adhere for 5 min before centrifuging the plate at 120 × *g* for 15 min at room temperature. The cells were fixed by adding prewarmed paraformaldehyde (PFA) directly into the culture media to a final dilution of 4% PFA, followed by centrifugation at 120 × *g* for 15 min. After three gentle washes with PBS, the cells were permeabilized with 0.1% Triton X-100 in PBS for 5 min and washed again thrice with PBS. Blocking was performed in 3% BSA and 5% donkey serum in PBS for 30 min. The cells were incubated with the primary antibodies (1:500 in blocking solution) overnight at 4 °C, and with Alexa 488- and 647-conjugated secondary antibodies (Invitrogen, 1:500 in blocking solution) for 2.5 h at room temperature. After every antibody incubation, the cells were washed three times with PBS-T (0.1% Tween-20 in PBS) for 5 min. DAPI (Thermo Scientific) was used to visualize nuclei. Images were acquired on a Leica SP8 confocal microscope using the ×63 objective (Leica Microsystems).

## Mice

Animal care and experimental protocols were in accordance with the Swiss Animal Protection Law and approved by the Veterinary Office of the Canton of Zurich (permits ZH131-16 and ZH188/2020). Mice were bred in a high hygienic grade facility of the University Hospital of Zurich (BZL) and housed in groups of 2–5. Mice were under a 12-h light/ 12-h dark cycle (from 7 a.m. to 7 p.m.) at 21 ± 1 °C. *Pycard*-deficient mice (B6.129-*Pycard*[tm1Vmd]) were generated as previously published in Mariathasan et al (2004).

                                                                    

C57BL/6 wild-type mice were obtained from the Jackson laboratory. To minimize environmental bias and potential differences in microbiota, littermates were bred. We performed the experiments with the highest possible gender and age congruence among experimental groups (Appendix Tables S5–S8). Mice were randomly assigned (unblinded) to the experiments. In the anti-ASC antibody treatment experiment two mice in each of the antibody-injected groups were found dead within the experiment.

## Genotype screening

Ear biopsies were digested and subjected to PCR. An 859 bp *Pycard* allele fragment was amplified using forward 5′-GAAGCTGCTGA-CAGTGCAAC-3′ and reverse 5′-CTCCAGGTCCATCACCAAGT-3′ primers. Amplification of a 275 bp gDNA fragment from the B6.129-*Pycard*<sup>tm1Vmd</sup> Neo cassette was done using forward 5′-TGGGACCAA-CAGACAATCGG-3′ and reverse 5′-TGGATACTTTCTCGGCAG-GAGC-3′ primers. PCR products were run on a 1.5% agarose gel and developed for genotype definition.

## AA induction and anti-ASC/isotype antibody injections

Silver nitrate (Merck) was eluted in nuclease-free water (Ambion®). AEF was prepared as previously described (Lundmark et al, 2002) and the pH value was adjusted to 7.4 before administration. AA amyloidosis induction was performed by injections of 100 µl AEF (i.v.) and 200 µl sterile-filtered 1% silver nitrate solution (s.c.) (Merck). Repeated injections of silver nitrate were in accordance with the reference (Vahdat Shariat Panahi et al, 2019). To assess amyloid load and ASC distribution in *Pycard*<sup>+/+</sup> and *Pycard*<sup>−/−</sup> mice AA was induced in 22 *Pycard*<sup>+/+</sup> and 24 *Pycard*<sup>−/−</sup> mice, and 33 mice were injected with either PBS, AgNO₃ or AEF and served as controls (Fig. 3A). To perform the anti-ASC antibody treatment, a total of 15 Pycard<sup>+/+</sup> mice were induced with AA (AgNO₃ and AEF injections, i.e., AA<sup>+</sup>) and additionally injected with either anti-murine ASC antibody or isotype control antibody (Fig. 4A). Anti-murine ASC and isotype control antibody were diluted in PBS with at a stock concentration of 2 mg/ml and intravenously injected at a concentration of 5 mg/kg body weight. Characteristics of experimental groups in Appendix Tables S5–S8. Two mice of the antibody-injected groups were found dead within the experiment and were therefore excluded from further analyses.

## Mouse serum amyloid A (SAA) measurements

To determine the SAA levels in mice serum, blood was withdrawn into BD Microtainer® SST™ Tubes at baseline and up to 96 h post injection. Samples were left at room temperature (RT) for 30 min and subjected to centrifugation at $10,000 \times g$ for 8 min at 4 °C. Serum samples were then transferred and stored at −80 °C. SAA levels were assessed by a mouse SAA enzyme-linked immunosorbent assay (ELISA) kit (abcam) according to the manufacturers' guidelines. Mouse serum samples were analyzed in technical triplicates. ELISA plate was developed and the optical density at 450 nm was measured.

## Euthanasia and organ harvesting

Upon euthanasia, organs were harvested and kept on ice in Iscove's Modified Dulbecco's Medium (IMDM) (ThermoFisher Scientific)

until measurement and further usage (no longer than 4 h). Bone marrow cells of tibia, femur and pelvis were flushed into IMDM using 25 G needles (B. Braun). Tissue was fixated in formalin for paraffin embedding or put in Tissue-Tek® O.C.T.™ compound (Sakura®) for frozen sections.

## Histology and immunohistochemistry of murine tissue

Formalin-fixed and paraffin-embedded spleen or cardiac sections (2–4 µm) were de-paraffinized with three cycles of xylene treatment followed by re-hydration with 100% EtOH, 96% EtOH, 70% EtOH, and water (each cycle 5 min) respectively. Hematoxylin and eosin (HE) as well as Congo Red (CR) staining and ASC immunohistochemistry (IHC) were performed according to standard procedures used at our institute. IHC was performed with anti-ASC pAb (clone AL177, AdipoGen) at a dilution of 1:500. To analyze the ASC-positive area (% of total area) the open-source software QuPath (https://qupath.github.io) was used. To perform LCP staining, slides were incubated with HS-310 for 30 min at RT in the dark at a final concentration of 0.3 µg/ml in PBS (Nilsson et al, 2010; Sjolander et al, 2016). After washing, slides were mounted with fluorescence mounting medium (Dako) and subjected to imaging on a fluorescence microscope (OLYMPUS BX61 fluorescence microscope system with an OLYMPUS XM10 camera). The hexameric LCP HS-310 was produced as previously described (Wahlström et al, 2020).

## Fluorescence and polarized microscopy

To analyze LCP-stained tissue sections, we assessed three different and independent visual fields at ×4 magnification and slides per mouse and organ. Two parameters, HS-310-positive area (% of area) as well as the fluorescence integrated density (A.U./µm²) of HS-310 were calculated with the open-source software ImageJ (https://github.com/imagej/ImageJ). To confirm the presence of amyloid, we assessed the apple-green birefringence of amyloid under polarizing light in Congo red-stained spleen tissue sections (Appendix Fig. S3). FACS-sorted splenic macrophages were visually confirmed by filter settings that allowed the detection of PE-Cy5 and APC-Cy7.

## SAA immunoblotting

To determine SAA presence by WB, spleen tissues from AA-induced mice were homogenized in 1:9 volumes (w/v) RIPA buffer (50 mM Tris pH 7.4, 1% NP-40, 0.25% Deoxycholic acid sodium salt, 150 nM NaCl, 1 mM EGTA, protease inhibitors (complete Mini, Roche)) using TissueLyser LT for 45 s for four cycles. Samples were cooled on ice between cycles. The supernatant was transferred into new tubes after full-speed centrifugation $16,000 \times g$ for 10 min at 4 °C. Input (same volume per sample and blot) was boiled at 95 °C for 10 min with a final concentration of 1 µM DTT and 4× NuPage™ LDS sample buffer (ThermoFisher Scientific). Spleen homogenate was separated using SDS-PAGE (Novex NuPAGE 4–12% Bis-Tris Gels, Invitrogen) and transferred to a PVDF membrane (ThermoFisher Scientific) at 20 V for 7 min. The membrane was then cut into two fractions, to allow an incubation with two distinct primary antibodies. The membrane was first blocked with 5% milk in TBS-T (Tris-Buffered Saline, 0.1%

TWEEN®20, pH 7.6) for 3 h at RT. Primary rabbit anti-SAA antibody (Invitrogen) was incubated overnight (o/n) at 4 °C at a concentration of 2.5 µg/ml (Appendix Table S9). In addition, the membrane was incubated with primary mouse anti-actin antibody (Merck) at a 1:8000 dilution in blocking buffer at 4 °C o/n. After four cycles of washing (10 min per cycle), membranes were incubated for 1 h at RT with secondary horseradish peroxidase (HRP)-conjugated goat anti-rabbit IgG (H + L) (Jackson Immuno) or with secondary HRP-conjugated goat anti-mouse IgG (H + L) (Jackson Immuno) diluted 1:5000 and 1:8000 in blocking buffer, respectively. Blots were developed using Luminata Crescendo Western HRP substrate (Millipore) and visualized with the Stella system (model 3200, Raytest) (Appendix Figs. S12 and S13).

## Production of monomeric ASC in *E. coli*

Monomeric human ASC-His, ASC-GFP-His and the ASC PYD domain (ASCPYD-His-SII) as well as the ASC CARD domain (ASCCARD-His-SII) were expressed in *E. coli* and purified from inclusion bodies via Nickel beads. Therefore, E. coli strain BL21(DE3) was transformed with pET-based vectors encoding the his-tagged ASC variants. Expression was done in self-inducing media (MagicMe-diaTM, Invitrogen) incubated at 37 °C for 1 h and 67 h at 20 °C. Subsequently, cells were harvested by centrifugation at 4000 rpm for 30 min. For cell lysis, the pellet was resuspended in 50 mM sodium phosphate, 300 mM NaCl, pH 7.5, and sonicated on ice 10 min at 40% power (2 sec pulse/pause). The suspension was then centrifuged at 14,000 $\times g$ at 4 °C for 30 min to collect the pellet containing the inclusion bodies. To solubilize the inclusion bodies, pellets were resuspended in 50 mM sodium phosphate, 300 mM NaCl, 6 M Guanidiniumhydrochloride (GdnHCl) 2 mM DTT, pH 7.5 for 30 min at RT. Afterward, the suspension was centrifuged at 14,000 $\times g$ at 4 °C for 30 min to remove residual insoluble cell debris. The supernatant was then incubated with Nickel beads (Themo, #88221) for 3 h at RT. The beads were washed once with 50 mM sodium phosphate, 300 mM NaCl, 6 M GdnHCl, 2 mM DTT, pH 7.5 and followed by 50 mM sodium phosphate, 300 mM NaCl, 6 M GdnHCl, 2 mM DTT, 20 mM imidazole pH 7.5. To elute the His-tagged proteins, beads were incubated with 50 mM sodium phosphate, 300 mM NaCl, 6 M GdnHCl, 2 mM DTT, 500 mM imidazole pH 7.5. The pH of the eluate was adjusted to pH 3.8 with diluted HCl. Next, the eluate was dialyzed against 50 mM Glycine, 150 mM NaCl, pH 3.8 in a MWCO 3500 Da cassette overnight (o/n) at 4 °C. To remove higher-order aggregates, the dialyzed samples were purified using a preparative size exclusion chromatography column (HiLoad 16/600 Superdex 75 pg). The monomeric proteins were finally concentrated using a VivaSpin MWCO 3000 Da spin column.

## Production and purification of ASC specks

Untagged ASC specks were recombinantly produced similarly as previously described (Martín-Sánchez et al, 2015). In brief, suspension HEK293 cells were transiently transfected with expression plasmids encoding full-length human ASC or ASC-GFP using linear 40 kDa polyethylenimine HCl (PEI). After 7 days of expression at 37 °C and 5% CO₂, cells were harvested and resuspended in Buffer A (320 mM sucrose, 20 mM HEPES-KOH (pH 7.5), 10 mM KCl, 1.5 mM MgCl₂, 1 mM, EDTA, 1 mM EGTA). Cells were lysed by syringing (10× 20 G,

20× 25 G), freeze-thawing (3×), followed by subsequent syringing (20× 25 G). Afterward, the lysate was centrifuged at $400 \times g$ for 8 min, the pellet was resuspended in 2 × CHAPS buffer (40 mM HEPES-KOH (pH 7.5), 10 mM MgCl₂, 1 mM EGTA, 0.2 mM PMSF, 0.2% CHAPS) and filtered using 5-µm centrifugal filters at $2000 \times g$ for 10 min. The filtrate was then diluted and gently mixed with 1 volume of 2 × CHAPS buffer and centrifuged at $2300 \times g$ for 8 min. The resulting pellet was resuspended in 1 ml of 1× CHAPS buffer and centrifuged at $5000 \times g$ for 8 min. This washing step was repeated twice. Afterward, the pellet was resuspended in 1× CHAPS buffer and loaded carefully on the top of 40% Percoll, and centrifuged at $16,000 \times g$ for 10 min. The interface layer containing the ASC speck particles was collected carefully and washed once by centrifugation at $5000 \times g$ for 3 min and resuspension in 1× CHAPS buffer. Lastly, tag-free ASC specks were stained using an anti-ASC antibody and a fluorescently labeled secondary antibody. Antibody-stained fluorescent particles were quantified in a fluorescence microscope using a Bürker chamber, delivering absolute counts of ASC speck numbers.

## ASC specks immunoblot analysis

To assess the purity and the presence of untagged ASC specks used in the aggregation assays, purified protein or HEK cell lysate were boiled at 95 °C for 10 min with a final concentration of 10 mM DTT diluted in NuPage™ LDS sample buffer (ThermoFisher Scientific). Proteins were separated using SDS-PAGE (Novex NuPAGE 4–12% Bis-Tris Gels, Invitrogen) and transferred to a PVDF membrane (ThermoFisher Scientific) at 20 V for 7 min. Membrane was blocked with 5% milk in TBS-T (Tris-Buffered Saline, 0.1% TWEEN®20, pH 7.6) for 3 h at RT. Primary rabbit anti-ASC antibody (clone AL177, AdipoGen) was incubated overnight (o/n) at 4 °C at a dilution of 1:1000. After 4 cycles of washing, membranes were incubated with secondary horseradish peroxidase (HRP)-conjugated goat anti-rabbit IgG (H + L) (Jackson Immuno) diluted 1:5000 in blocking buffer for 1 h at RT. The membrane was developed as described above (Appendix Fig. S14).

## In vitro SAA fibril formation

Recombinant murine SAA1 was generated as previously described in (Claus et al, 2017). The in vitro aggregation assay was carried out in a black 96-well plate (Greiner Bio-One, PS, F-bottom, black) on a FLUOstar OMEGA plate reader (BMG Labtech). SAA1 was dissolved in water to achieve a stock solution concentration of 10 mg/ml. The final reaction volume was 100 µl per well and consisted of 50 µM murine SAA1, 20 µM Thioflavin T (abcam) and 50 mM Tris buffer-HCl, pH 8.0. ASC specks (quantified as described above) were diluted in PBS and added at various concentrations. Bovine serum albumin (BSA, Sigma-Aldrich) was dissolved in water to reach a final concentration of 50 µM. The plate was incubated at 37 °C and agitated every 20 min by orbital shaking for 10 s at 100 rpm. Fluorescence (Ex: 450 nm, Em: 490 nm) was measured over the course of 150 h. The control and dose-response experiments were performed independently. ThT fluorescence intensity values were displayed in the following manner: (1) We first transformed the data so that all conditions and repeats started with ThT fluorescence intensity 0 at timepoint 0 min. (2) We applied a range function separately onto both the

control experiment as well as on the dose-response datasets, i.e.,

$$\frac{x - \min(x)}{\max(x) - \min(x)}$$

where $x$ was the adjusted ThT fluorescence intensity and min and max the respective minimum and maximum ThT fluorescence signals in each of the two experiments. This allowed us to obtain normalized datasets scaled between 0 and 1 (3) To visualize trends over all replicates, we conducted four-parametric logistic regression analyses. (4) For quantification of $t_{1/2}$ values, data were fitted with a global fitting procedure using the software application AmyloFit according to the developer's instructions (Meisl et al, 2016).

## Limited proteolysis-coupled mass spectrometry (LiP-MS)

LiP-MS was conducted as shown earlier (Cappelletti et al, 2021; Feng et al, 2014; Pepelnjak et al, 2020; Schopper et al, 2017; Schurch et al, 2022). Briefly, we incubated purified recombinants ASC protein or ExpiHEK cell native lysates containing ASC specks with human recombinant SAA1 proteins for 15 min at 37 °C in LiP buffer (100 mM HEPES pH 7.4, 150 mM KCl, 1 mM MgCl$_2$). Next, PK was added to each independent technical replicate simultaneously at 1:100 (w/w) enzyme to substrate ratio for 5 min at 37 °C. Four technical replicates per condition were done for purified proteins and three for ExpiHEK lysate. To stop the limited proteolysis reaction, PK was heat-inactivated by sample incubation at 99 °C. Subsequently, samples were transferred into equal volume of 10% sodium deoxycholate (Sigma-Aldrich). Next, samples were reduced with 5 mM tris (2-carboxyethyl)phosphine hydrochloride for 40 min at 37 °C under 800 rpm shaking, alkylated in 40 mM iodoacetamide, and incubated in the dark at RT for 30 min. Finally, samples were diluted in ammonium bicarbonate and digested with lysyl endopeptidase and trypsin (at enzyme to substrate ration of 1:100) at 37 °C for 17 h under 800 rpm shaking. Digestion was stopped by the addition of formic acid (4% final concentration). Precipitate of sodium deoxycholate was removed via centrifugation, samples were desalted with Sep-Pak tC18 cartridges (Waters) and eluted with 80% acetonitrile, 0.1% formic acid. Samples were analyzed on an Orbitrap Fusion Lumos Tribrid mass spectrometer (ThermoFisher) equipped with a nanoelectrospray ion source and an UPLC system (Waters) in data-independent acquisition mode. Spectronaut (Biognosys AG) software was used to search the raw MS data. The statistical data analysis was done in R. Two-tailed $t$ test was applied to assess statistical significance between peptide abundances. Significance cutoffs of $|\log_2$ (fold change)$|>1$ and $-\log_{10}$ (FDR-adjusted $P$ value) $<0.05$ or $|\log_2$ (fold change)$|>1$ and $-\log_{10}$ ($P$ value) $<0.05$ were used, as indicated.

## Differentiation of bone marrow cells to bone marrow-derived macrophages (BMDMs)

BMDMs were generated, with slight modifications, as previously published (Yanagita et al, 2017). BM cells were harvested from the tibia and femur, washed twice in PBS, and resuspended in differentiation medium I–10 + M-CSF (i.e., IMDM, 10% FBS, 1 mg/ml Pen/Strep containing 25 ng/ml recombinant murine M-CSF (PeproTech)) at a density of 1–2 Mio. cells/ml. After

incubation for 4 days at 37 °C differentiation medium was exchanged. On day 8, attached BMDMs were detached using Accutase® (Innovative Cell Technologies, Inc.), washed twice in PBS, counted, and used in the phagocytosis assay.

## Anti-ASC antibody cloning

The anti-ASC antibody was cloned from B-cells of rabbits immunized with recombinant ASC. Isotype control antibody is a similarly obtained anti-idiotypic rabbit monoclonal antibody isolated from a rabbit immunized with a human IgG1 Fab. Recombinant antibodies were produced in a transient HEK expression system and purified via protein A affinity purification on an automated AKTA system (AKTA explorer FPLC, GE Healthcare). For the in vivo studies, rabbit V-regions were fused with the murine constant region of the mouse IgG2a heavy chain and kappa chain to minimize immunogenicity. Recombinant antibodies were produced as described above but in addition remaining endotoxin was removed by hydrophobic interaction chromatography (AKTA explorer FPLC, GE Healthcare). Endotoxin levels were measured (Endosafe nexgen, Charles River) and only antibodies with less than 0.15 EU/mg were used for the in vivo studies.

## Anti-murine ASC antibody binding assay

High-binding clear flat bottom 384-well plate (Huber lab) were coated o/n (4 °C) with 10 μl/well of murine ASC (Cusabio, CSB-EP861664 MO) at a concentration of 1.25 μg/ml or BSA at a concentration of 1 μg/ml diluted in PBS. After o/n incubation, wells were aspirated and washed once with 100 μl/well of wash buffer (0.05% Tween®20 in PBS). Wells were blocked for an hour with 50 μl of 2% BSA diluted in PBS at RT. After blocking, wells were incubated 2 h at RT with 10 μl of test antibody (anti-ASC or isotype ctrl) and serially diluted. Wells were washed four times and consecutively incubated with 10 μl/well of detection antibody for an hour at RT. Unbound detection antibody was removed by washing four times, and the wells were incubated with 20 μl TMB for 5 min. The reaction was stopped by adding 10 μl of 1 M H$_2$SO$_4$ per well. Absorbance was immediately read at 450 nm.

## Anti-ASC antibody serum concentration measurements

Serum was isolated as previously described. 384-well SpectraPlate HB plate was coated with recombinant ASC-his at a concentration of 1 μg/ml in PBS (pH 7.4) and incubated o/n (4 °C) with 20 μl/well. Plates were then washed three times with sample buffer (0.1% Tween®20 in PBS) and blocked with blocking buffer (5% SureBlockTM (Lubio Science, #SB232010-250G) in sample buffer) for 90 min at RT. The blocking buffer was removed by plate flipping. Murine serum samples as well as positive (anti-ASC monoclonal ab, Mabylon AG) and negative (isotype control ab, Mabylon AG) assay controls were resuspended in sample buffer followed by serial dilution and incubation for 2 h at 37 °C. The starting concentration of anti-ASC mAb was 2 μg/ml. Starting dilution of mouse serum samples was 1:20. After four wash cycles, HRP-conjugated secondary antibody (Goat anti-mouse IgG (H + L), Jackson ImmunoResearch, #115-035-003) were incubated for 1 h, at RT at a dilution of 1:4000 (in sample buffer). After three

wash cycles, plate was developed with 20 µl/well TMB and incubated for 5 min at RT in the dark. The reaction was stopped with 20 µl/well 0.5 M $H_2SO_4$. Finally, absorbance was read at $\lambda = 450$ nm using an EnVision plate reader (PerkinElmer). Serum samples as well as controls were assayed in technical duplicates.

## In vitro phagocytosis

The phagocytosis of BMDMs was performed using a phagocytosis assay kit (abcam). Individual samples were prepared in duplicates. The in vitro assay was performed in 96-well tissue culture plates (TPP). *Pycard*[+/+] and *Pycard*[-/-] BMDMs were suspended at a density of 500,000 cells/ml. BMDMs were stimulated, with slight modifications, as previously published (Chen et al, 2020). BMDMs were stimulated with murine SAA1 at a concentration of 2 µg/ml for 16 h at 37 °C prior to phagocytosis substrate encounter. After 1 h incubation with the phagocytosis substrate at 37 °C, cells were treated according to the manufacturer's guidelines and proceeded to EnVision Multimode Plate Reader (PerkinElmer) for data acquisition. The absorbance was determined at 405 nm.

## Assessment of cellular spleen composition by flow cytometry

For the analysis of the spleen composition, cells were isolated from spleens using a 70 µm cell strainer (Falcon®). Following red blood cell lysis, cells were stained with the following monoclonal antibodies to determine the cellular architecture of most abundant spleen and immune cells (Appendix Table S10): anti-mouse CD45.2 (Biolegend), anti-mouse Gr-1 (eBioscience), anti-mouse F4/80 (eBioscience), anti-mouse CD11b (Biolgegend), anti-mouse B220 (Biolegend), anti-mouse CD3 (Biolegend), anti-mouse CD206 (Biolegend), anti-mouse MHCII (Biolegend) and anti-mouse CD11b (eBioscience). Data acquisition and analysis were performed on a BD LSRIIFortessa™ flow cytometer and FlowJo v10.6.1, respectively. Gating strategy was applied, with slight modifications, as previously published (Arlt et al, 2020; Nawaz et al, 2017; Ono et al, 2018).

## RNA extraction and high-throughput sequencing (NGS)

Upon flow cytometric sorting of F4/80 + /CD11b+ splenic macrophages originating from AA[+] mice (see above), total RNA was extracted using TRIzol™ Reagent (ThermoFisher Scientific) according to the manufacturer's guidelines and proceeded to high-throughput sequencing (NGS). RNA integrity and quantity were assessed by RNA ScreenTape Analysis (Agilent). The SMARTer Stranded Total RNA-Seq Kit-Pico Input Mammalian (Takara) together with a NovaSeq platform (Illumina) was applied for transcriptomic data acquisition. Data was analyzed using established analysis pipelines at the Functional Genomics Center Zurich (FGCZ).

## Complete blood count (CBC) from peripheral blood

Blood was withdrawn into Microvette® 100 K3E (SARSTEDT Germany) cuvettes according to the manufacturer's guidelines. To assess CBCs, samples were run on an ADVIA (Siemens Healthineers) hematology system.

## Experimentation with human samples

All experiments and analyses involving samples from human donors were conducted with informed consent and the approval of the ethics committee of the Canton Zurich (KEK-ZH-Nr. 2015-0561, BASEC-Nr. 2018-01042, and BASEC-Nr. 2022-00293) and in accordance with the provisions of the Declaration of Helsinki and the Good Clinical Practice guidelines of the International Conference on Harmonization.

## High-throughput antibody profiling in unselected patient cohort and validation experiments

Serological screens were conducted as reported (Emmenegger et al, 2022; Emmenegger et al, 2021b; Senatore et al, 2020). Briefly, high-binding 1536-well plates (PerkinElmer, SpectraPlate 1536 HB) were coated with 1 µg/ml human ASC protein containing a C-terminal his-tag in PBS at 37 °C for 1 h, followed by three washes with PBS-T and by blocking with 5% milk in PBS-T for 1.5 h. Three microliter plasma, diluted in 57 µl sample buffer (1% milk in PBS-T), were dispensed at various volumes into human ASC-coated 1536-well plates using contactless dispensing with an ECHO 555 Acoustic Dispenser (Labcyte). Thereby, dilution curves ranging from plasma dilutions 1:50 to 1:6000 were generated (eight dilution points per patient plasma sample). After the sample incubation for 2 h at RT, the wells were washed five times with wash buffer and the presence of IgGs bound to ASC were detected using an HRP-linked anti-human IgG antibody (Peroxidase AffiniPure Goat Anti-Human IgG, Fcγ Fragment Specific, Jackson, 109-035-098, at 1:4000 dilution in sample buffer). The incubation of the secondary antibody for one hour at RT was followed by three washes with PBS-T, the addition of TMB, an incubation of three minutes at RT, and the addition of 0.5 M $H_2SO_4$. The well volume for each step reached a final of 3 µl. The plates were centrifuged after all dispensing steps, except for the addition of TMB. The absorbance at 450 nm was measured in a plate reader (PerkinElmer, EnVision) and the inflection points of the sigmoidal binding curves were determined using a custom-designed fitting algorithm. Samples reaching half-maximum saturation (shown as the inflection point of the logistic regression curve) at a concentration ≤1:100, i.e., at $p(EC_{50}) \geq 2$, and with a mean squared residual error <20% of the actual $p(EC_{50})$ were considered hits. The inclusion of a threshold for fitting error ensures a reliable identification of positives from high-throughput screening. Negative $p(EC_{50})$ values, reflecting nonreactive samples, were rescaled as zero.

For the validation screen, hits from the high-throughput screen were tested against a panel of antigens consisting of recombinant human ASC protein, human recPrP[23-230], the full-length tau protein (Tau441), the SARS-CoV-2 Spike ectodomain and natural Ara h 2 allergen. The validation screen was performed identically to the primary high-throughput screen, except that duplicates were used instead of unicates and a 384-well plate format was chosen. Samples from the validation screen were considered confirmed if $p(EC_{50})$ for PrP ≥2 (distinct reactivity against ASC) and $p(EC_{50})$ for other targets <2 (no distinct reactivity against any other control target), except for the SARS-CoV-2 Spike protein, which was included to assess potential associations with acute infection or vaccination. The following assay-positive controls were used at a starting concentration of 1 µg/ml: (1) Anti-ASC/TMS1/PYCARD

Antibody (B-3) mouse monoclonal IgG1 (sc-514414, Santa Cruz); (2) anti-human PrP antibody huPOM1 (Senatore et al, 2020); (3) Anti-Tau (4-repeat isoform RD4) Antibody, clone 1E1/A6 (05-804, Sigma-Aldrich). Additional assay-positive controls used at a starting dilution of 1:50 were: a plasma pool of patients admitted to the University Hospital of Zurich due to COVID-19 (anti-Spike PC); plasma of a patient identified to be seropositive for IgG against nAra h 2 (anti-Ara h 2 PC). The experiment on the two ASC domains (PYD and CARD, both of them containing a C-terminal his-tag; produced by Mabylon AG, Schlieren) and on additional his-tagged proteins as controls (his-LAG3 (Emmenegger et al, 2021a), his-TIM3 (AcroBiosystems)) was conducted using the same ELISA protocols as outlined above. Antigens and antibodies used for the high-throughput antibody profiling are shown in Appendix Tables S11 and S12.

## Statistical analysis

Statistical analysis was performed using GraphPad Prism v9, Python3, and R. If not indicated otherwise, we conducted the Kruskal–Wallis test with post-hoc Wilcoxon rank-sum test with Holms correction for multiple comparison on non-parametric data. Statistical details are described in the respective figure legends and in the Result section. A two-tailed $P$ value $< 0.05$ was considered as statistically significant in all group-based experiments, except for the exploratory ICD-10 code-based analysis where the significant threshold, $\alpha$, was 0.01. Confidence intervals were calculated at a confidence level of 95%. When reporting medians, we usually provide the interquartile range (IQR). Human data are stored in an *MS-SQL* database. Multiple regression with a logit link function using various p($EC_{50}$) cutoff values for ICD-10 data exploration was performed using the rstanarm package (https://mc-stan.org/rstanarm/) with a Bayesian LASSO prior, similar to what was expounded in detail previously (Emmenegger et al, 2023a; Emmenegger et al, 2023b; Lamparter et al, 2022).

## Data availability

RNA-seq data: The RNA sequencing data for this study have been deposited in the European Nucleotide Archive (ENA) at EMBL-EBI under accession number PRJEB76882. https://www.ebi.ac.uk/ena/browser/view/PRJEB76882. LiP-MS data: The mass spectrometry proteomics data for this study have been deposited to the ProteomeXchange Consortium via the PRIDE (Perez-Riverol et al, 2022) partner repository with the dataset identifier PXD053372. https://www.ebi.ac.uk/pride/markdownpage/pridesubmissiontool. Confocal/STED microscopy images: Imaging data for this study have been deposited on BioImage Archive under the accession number S-BIAD1248. https://www.ebi.ac.uk/biostudies/bioimages/studies/S-BIAD1248. Primary reagents can be shared and made available upon request. Antibodies against ASC are the property of Mabylon AG and will be shared upon reasonable request under specific material transfer agreement terms.

The source data of this paper are collected in the following database record: biostudies:S-SCDT-10_1038-S44321-024-00107-0.

**The paper explained**

**Problem**

Upon activation of ASC-containing inflammasomes, intracellular molecular assemblies-ASC specks-are released from myeloid cells into the extracellular space, where they cross-seed the aggregation of Aβ amyloid in Alzheimer's disease. This raised the question whether the ASC inflammasome adapter is involved in additional aggregation proteinopathies, such as inflammation-induced amyloid A (AA) amyloidosis.

**Results**

In this study, we report that ASC governs the extent of inflammatory (AA) amyloidosis, a systemic disease caused by the aggregation, peripheral deposition and invasion of the acute-phase reactant serum amyloid A (SAA). A large-scale anti-ASC autoantibody screening of 23,450 plasma samples from 19,334 hospital patients revealed rare immunoreactivity towards ASC and was not associated with any specific disease, indicating that anti-ASC antibody treatment modalities would not be confounded by natural autoimmunity.

**Impact**

These findings expand the role played by ASC and IL-1 independent inflammasome employments to extraneural proteinopathies of humans and experimental animals and suggest that anti-ASC immunotherapy may contribute to resolving inflammatory diseases and their sequelae.

## Peer review information

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

## Acknowledgements

The authors appreciate that Dr. Emmanuel Contassot shared the B6.129-*Pycard*tm1Vmd mice. We thank Dr. Marcus Fändrich and his colleagues for providing murine SAA1 protein. The authors are grateful that Dr. Itzel Condado-Morales and Dr. Sathish Kumar provided technical support with the aggregation assay and data analysis. The authors thank Dr. Markus Manz that the authors could use the ADVIA hematology system. The authors thank Inga Hochheiser and Dr. Matthias Geyer (ISB, University of Bonn, Germany) for providing the human ASC and the team of Mabylon AG (Schlieren) for sharing the ASC-PYD and the ASC-CARD proteins. The authors thank Rita Moos, Jingjing Guo, Mirzet Delic and Paulina Pawlak for their valuable technical knowledge, advice and assistance in mouse breeding and experimental injections. The authors thank the histology team at the Institute for Neuropathology for their support. The authors thank the RNA sequencing core from the University of Zurich for their help in the data analysis. The high-throughput antibody profiling-related work was funded through an SPHN driver grant (2017DRI1) and through NOMIS Foundation, the Schwyzer Winiker Stiftung, and the Baugarten Stiftung (coordinated by the USZ Foundation, USZF27101) awarded to AA and ME. ML is supported by the Swiss National Science Foundation (grant number 210931). AA is supported by the Swiss National Science Foundation (grant number 179040) and the NOMIS Foundation.

## Author contributions

**Marco Losa**: Conceptualization; Resources; Data curation; Formal analysis; Funding acquisition; Validation; Investigation; Visualization; Methodology; Writing—original draft; Project administration; Writing—review and editing. **Marc Emmenegger**: Resources; Data curation; Software; Formal analysis; Funding acquisition; Validation; Investigation; Visualization; Methodology; Writing—original draft; Project administration; Writing—review and editing. **Pierre De Rossi**: Data curation; Formal analysis; Validation; Visualization; Writing—review and editing. **Patrick M Schürch**: Resources; Data curation; Visualization; Methodology; Writing—review and editing. **Tetiana Serdiuk**: Resources; Data curation; Software; Formal analysis; Validation; Investigation; Visualization; Methodology; Writing—review and editing. **Niccolò Pengo**: Resources; Data curation; Methodology. **Danaëlle Capron**: Resources; Data curation; Methodology. **Dimitri Bieli**: Resources; Data curation; Methodology. **Niklas Bargenda**: Resources; Data curation; Methodology; Writing—review and editing. **Niels J Rupp**: Resources. **Manfredi C Carta**: Data curation; Methodology. **Karl J Frontzek**: Resources; Methodology. **Veronika Lysenko**: Resources; Data curation; Methodology. **Regina R Reimann**: Resources; Data curation. **Petra Schwarz**: Resources; Project administration. **Mario Nuvolone**: Resources; Methodology; Project administration. **Gunilla T Westermark**: Resources; Methodology. **K Peter R Nilsson**: Resources; Methodology. **Magdalini Polymenidou**: Resources; Methodology. **Alexandre PA Theocharides**: Resources. **Simone Hornemann**: Resources; Data curation; Methodology; Writing—review and editing. **Paola Picotti**: Resources; Methodology. **Adriano Aguzzi**: Conceptualization; Resources; Supervision; Funding acquisition; Visualization; Methodology; Writing—original draft; Project administration; Writing—review and editing.

Source data underlying figure panels in this paper may have individual authorship assigned. Where available, figure panel/source data authorship is listed in the following database record: biostudies:S-SCDT-10_1038-S44321-024-00107-0.

## Disclosure and competing interests statement

Adriano Aguzzi is a founder, shareholder, director, and consultant for Mabylon AG, which has developed the anti-ASC monoclonal antibody used in this study. NP, DC and DB are employees of Mabylon AG. Adriano Aguzzi is a member of the Advisory Editorial Board of EMBO Molecular Medicine. This has no bearing on the editorial consideration of this article for publication.

