## [Peer Review File · EMBO Molecular Medicine]

The ASC inflammasome adapter governs SAA-derived protein aggregation in inflammatory amyloidosis

Marco Losa, Marc Emmenegger, Pierre De Rossi, Patrick Schuerch, Tetiana Serdiuk, Niccolo Pengo, Danaelle Capron, Dimitri Bieli, Niklas Bargenda, Niels Rupp, Manfredi Carta, Karl Frontzek, Veronika Lysenko, Regina Reimann, Petra Schwarz, Mario Nuvolone, Gunilla Westermarck, Peter Nilsson, Magdalini Polymenidou, Alexandre Theocharides, Simone Hornemann, Paola Picotti, and Adriano Aguzzi

Corresponding author: Adriano Aguzzi (adriano.aguzzi@usz.ch)

Review Timeline:

Transfer from Review Commons:	28th May 24
Editorial Decision:	18th Jun 24
Revision Received:	5th Jul 24
Accepted:	10th Jul 24

Editor: Jingyi Hou

Transaction Report:

Review
COMMONS

This manuscript was transferred to EMBO Molecular Medicine following peer review at Review Commons.

Review #1

1. Evidence, reproducibility and clarity:

Evidence, reproducibility and clarity (Required)

In this manuscript, et al., investigates the role of the inflammasome adapter ASC (in AA amyloidosis). This condition involves the aggregation of serum amyloid A (SAA) and is linked to chronic inflammation.

Firstly, I can directly say that I do recommend this study for publication. This is a well conducted and well-written study which advances the knowledge on IL-1-independent inflammatory functions of inflammasomes. Furthermore, I find it particularly impressive that despite the inflammasome research community is well aware that amyloidosis is a hallmark of inflammatory diseases, it took a neuroscientist specialized in prion diseases to raise the question whether ASC would be involved in seeding serum AA aggregation.

Key findings include:

- ASC forms extracellular aggregates that enhance SAA aggregation, as observed through superresolution microscopy.
- In a mouse model, the absence of ASC significantly reduced amyloid load, not due to increased phagocytosis but likely due to diminished aggregation.
- Treatment with anti-ASC antibodies reduced amyloid load and mitigated weight loss in mice with AA amyloidosis.

These findings suggest that ASC plays a crucial role in AA amyloidosis and that targeting ASC could be a potential therapeutic strategy. The study expands our understanding of the involvement of ASC in proteinopathies beyond neural diseases, pointing to its role in systemic conditions like AA amyloidosis.

Main Comments:

Overall, the experiments are well-conducted and mostly all controls I would expect were included. With few exceptions, the data is convincing. With that said, I have issues with some of the staining employed in Fig 1.

In Fig. 1, the authors assess ASC staining in cardiac tissues from a patient with vasculitis and systemic inflammation-related AA amyloidosis, and a control patient who died of a heart attack but had no signs of amyloidosis. However, most of the data shown is related to the AL177 anti-ASC. More importantly, no isotype stainings are included. We have previously demonstrated that the AL177 anti-ASC, used here, reacts quite strongly with ASC^{-/-} cells, and it is one of the less specific anti-ASC commercially available (PMID: 27221487). As this is data from one patient (understandably), I wonder if the authors could counterstain ASC in the same samples using a specific human anti-ASC with a different color (ex: Biolegend HASC), and confirm that the signal overlays with the AL-177.

Finally, in Figure 1H it seems from the description that another anti-ASC was used: "referred in the legend as ASC (MAB ASC, Yellow)". Is this a monoclonal anti-ASC? Also, the images show large and bright antibody aggregates (middle of the image, top left corner behind the "H", and a massive fluorescence in the bottom right of the image), indicating the presence of staining artifacts. Again, no counterstaining with isotype controls are shown.

Overall, although I don't dispute the possibility that ASC would co-localize with SAA deposits, I don't think the data presented can safely sustain that claim. I would, therefore, suggest that alternative methods to be employed to substantiate these conclusions: Supposedly, would it be possible to immuno-precipitate (IP) amyloid SAA and assess ASC via western blotting? As well as IP ASC and detect SAA? Or use DSS-crosslinking to find ASC oligomers in tissue areas rich in SAA?

****Minor comments:****

In addition to these main comments, some minor adjustments are recommended:

For example, it would be reasonable to quantify the results in Figure 3G and providing clarification regarding the controls in the figure legend. Though there is significantly less SAA in spleen homogenates from *Asc*^{-/-}, there also seems to be the case for b-actin in Fig 3G. Moreover, in the figure legend the authors state: "...Spleen homogenate from untreated (-ctrl) and AA+ (+ctrl) C57BL/6 wt mice from an independent experiment served as negative and positive control, respectively."

I don't know what the authors mean with that. Is this a montage, or samples from different experiments were run together in one blot? And if so, for what reason? This is confusing and should be clarified.

Furthermore, in the Abstract, a slight rephrasing is suggested to accurately describe ASC specks as molecular aggregates formed inside cells, which are subsequently released into the extracellular space.

Lastly, enhancing the text size in figures, particularly in Fig 3, is advised to improve legibility and overall clarity.

2. Significance:

Significance (Required)

In conclusion, this manuscript offers valuable insights into the role of ASC in AA amyloidosis, presenting compelling findings that support its potential as a therapeutic target. Addressing the mentioned concerns and making the suggested revisions will further enhance the manuscript's scientific rigor and impact. Overall, this study is a valuable contribution to the field of inflammasome research and its relevance in systemic conditions like AA amyloidosis.

3. How much time do you estimate the authors will need to complete the suggested revisions:

Estimated time to Complete Revisions (Required)

(Decision Recommendation)

Between 1 and 3 months

Yes

Review #2

1. Evidence, reproducibility and clarity:

Evidence, reproducibility and clarity (Required)

The manuscript by Losa et al., investigates whether ASC is involved in serum AA amyloidosis. The authors report that ASC colocalizes with SAA in human AA amyloidosis and that purified ASC specks accelerate SAA fibril formation in vitro. In addition, splenic AA amyloid was decreased in Pycard^{-/-} mice compared to Pycard^{+/+} mice and that treatment with anti-ASC antibodies decreased amyloid loads in Pycard^{+/+} mice. Lastly, they analyzed serum of 19,334 patients to show that the prevalence of anti-ASC antibodies did not correlate with any specific disease. The authors conclude that ASC to play a role in extraneural proteinopathies of humans and experimental animals and suggest that anti-ASC immunotherapy may contribute to resolving such diseases. The findings in the study are novel and demonstrate a new role for ASC in aggregation proteinopathies. However, there are number of issues that need to be addressed before acceptance for publication.

****Major Points:****

Figure 3 E depicts Western blots of monomeric SAA in spleen of Pycard^{+/+} and Pycard^{-/-} mice. The authors should include immunoblots depicting the levels of ASC in these tissues and to demonstrate that the Pycard^{-/-} mice lack ASC. Fig. 3B shows that at 96 hours after injection there was no difference in SAA serum concentration. How do the authors explain this drop in SAA serum concentration? No explanation is provided.

Figure 4 shows anti-ASC administration reduces amyloid load. The immunoblot in Figure 4C does not represent the quantification of the blot. In fact, there are only 3 samples per treatment group whereas the quantification shows 5-6 animals per group. Additionally, the authors have not shown that the drug penetrates the target tissue and how much drug is present in spleen to provide a therapeutic effect. What is the half-life of the drug? These parameters are critical to assess the MOA of the anti-ASC used in these studies.

The authors should expand the discussion section to include the work of other groups that have successfully employed anti-ASC antibodies. For example, PMID: 35793783, PMID: 32366256

Methods: The authors provide the number of animals employed in the Supplemental Tables 5 and 7. These numbers should be provided in the methods section or in the Figure legends. Additionally, how many replicates were performed for the data in Figure 2?

2. Significance:

Significance (Required)

The findings in the study are novel and demonstrate a new role for ASC in aggregation proteinopathies. This study reports a crucial role for ASC in SAA interaction and recruitment, SAA serum level modulation, SAA fibril formation acceleration, and controlling the extent of inflammation associated amyloidosis with respect to AA amyloid deposition

3. How much time do you estimate the authors will need to complete the suggested revisions:

Estimated time to Complete Revisions (Required)

(Decision Recommendation)

Less than 1 month

4. Review Commons values the work of reviewers and encourages them to get credit for their work. Select 'Yes' below to register your reviewing activity at Web of Science

Reviewer Recognition Service (formerly Publons); note that the content of your review will not be visible on Web of Science.

Yes

Review #3

1. Evidence, reproducibility and clarity:

Evidence, reproducibility and clarity (Required)

The manuscript by Losa et al. explores the co-aggregation of ASC with serum amyloid A (SAA) in vivo and in mouse models, It posits that, similar to Amyloid beta, SAA is cross-seeded by ASC foci both in vitro and in vivo. This review only addresses the co-localization and in vitro cross seeding data (Figs. 1 and 2A, B), not the mouse experiments or mass spectrometry data.

The manuscript first shows co-deposition of ASC with SAA amyloid. SAA was stained both with Congo red and ThS, both standard dyes for amyloid staining. Figure S2 shows CR birefringence, the hallmark of amyloid deposits. The authors then move to demonstrate co-localization of SAA and ASC in confocal and STED immuno-fluorescence microscopy.

Confocal images C-E show overlapping staining of markers for both SAA and ASC. Similarly, STED images show co-aggregation of ASC and SAA in amyloidosis patients. However, since confocal images F and G seem to show overlapping staining of the yellow and magenta channels as well, a careful quantitative analysis of the data I needed. Quantify co-localization (Pearson coefficient) in confocal and STED images. STED images from control patients are missing and need to be included.

The authors then move on to demonstrate that ASC foci can cross-seed SAA amyloid formation in vitro, by recording SAA aggregation kinetics in the presence and absence of ASC foci. Curves recorded in the presence of ASC foci have accelerated kinetics as shown by a decrease in the time to reach half-maximal fluorescence ($t_{1/2}$). However, these data (Fig 2A, B) are not very clean. Only three data points out of five curves shown in panel A. are presented in the fitting of the control (yellow) aggregation kinetics in panel B. Why was this done? Panel B shows a significant difference between the control and the kinetics seeded with ASC specks. It looks doubtful that the results are still statistically significant if these data are included, so their exclusion impacts the overall conclusion of the paper. The significance of the cross-seeding results needs to be substantiated experimentally.

2. Significance:

Significance (Required)

The discovery of the role of ASC in Alzheimer's disease generated an exciting new hypothesis to the etiology of sporadic AD, for which the cause is unknown. The current manuscript finds that ASC may also play a role in AA amyloidosis, which is a significant finding.

3. How much time do you estimate the authors will need to complete the suggested revisions:

Estimated time to Complete Revisions (Required)

(Decision Recommendation)

Between 1 and 3 months

Yes

Full Revision

Manuscript number: RC-2023-02235

Corresponding author(s): Adriano, Aguzzi

1. General Statements

We thank the reviewers for providing valuable comments. We are pleased that our study is considered important to advance the knowledge on IL-1-independent inflammatory functions of inflammasomes. We have clarified and revised the manuscript (track changed) as detailed below in the point-by-point response in this letter.

Referee 1

General: In this manuscript, et al., investigates the role of the inflammasome adapter ASC (in AA amyloidosis). This condition involves the aggregation of serum amyloid A (SAA) and is linked to chronic inflammation.

Firstly, I can directly say that I do recommend this study for publication. This is a well conducted and well-written study which advances the knowledge on IL-1-independent inflammatory functions of inflammasomes. Furthermore, I find it particularly impressive that despite the inflammasome research community is well aware that amyloidosis is a hallmark of inflammatory diseases, it took a neuroscientist specialized in prion diseases to raise the question whether ASC would be involved in seeding serum AA aggregation.

Key findings include:

- ASC forms extracellular aggregates that enhance SAA aggregation, as observed through superresolution microscopy.
- In a mouse model, the absence of ASC significantly reduced amyloid load, not due to increased phagocytosis but likely due to diminished aggregation.
- Treatment with anti-ASC antibodies reduced amyloid load and mitigated weight loss in mice with AA amyloidosis.

These findings suggest that ASC plays a crucial role in AA amyloidosis and that targeting ASC could be a potential therapeutic strategy. The study expands our understanding of the involvement of ASC in proteinopathies beyond neural diseases, pointing to its role in systemic conditions like AA amyloidosis.

Significance: In conclusion, this manuscript offers valuable insights into the role of ASC in AA amyloidosis, presenting compelling findings that support its potential as a therapeutic target. Addressing the mentioned concerns and making the suggested revisions will further enhance the manuscript's

scientific rigor and impact. Overall, this study is a valuable contribution to the field of inflammasome research and its relevance in systemic conditions like AA amyloidosis.

Comment 1: Overall, the experiments are well-conducted and mostly all controls I would expect were included. With few exceptions, the data is convincing. With that said, I have issues with some of the staining employed in Fig 1. In Fig. 1, the authors assess ASC staining in cardiac tissues from a patient with vasculitis and systemic inflammation-related AA amyloidosis, and a control patient who died of a heart attack but had no signs of amyloidosis. However, most of the data shown is related to the AL177 anti-ASC. More importantly, no isotype stainings are included. We have previously demonstrated that the AL177 anti-ASC, used here, reacts quite strongly with ASC^{-/-} cells, and it is one of the less specific anti-ASC commercially available (PMID: 27221487). As this is data from one patient (understandably), I wonder if the authors could counterstain ASC in the same samples using a specific human anti-ASC with a different color (ex: Biolegend HASC), and confirm that the signal overlays with the AL-177.

Response: We conducted additional experiments to address the anti-ASC antibody specificity, as now described in Results, Method, and Fig. S1. We tested a set of anti-ASC antibodies (AL177, MY6745, 1C3D7) for their ASC specificity. We confirmed that both the AL177 and the MY6745 antibodies have high ASC-specificity (Fig. S1A). Moreover, for illustration purposes (and to warn other scientists), we included a third anti-ASC antibody (1C3D7) found to be unspecific as it yielded a strong signal in *PYCARD*^{-/-} (ASC^{-/-}) THP-1 cells (Fig. S1B). In addition, isotype controls were included in these experiments (Fig. S1A, right panels), as suggested by the reviewer, showing no target protein detection in both, *PYCARD*^{+/+} (ASC^{+/+}) and *PYCARD*^{-/-} cells underscoring the anti-ASC specificity of AL177 and MY6745 antibodies.

Comment 2: Finally, in Figure 1H it seems from the description that another anti-ASC was used: "referred in the legend as ASC (MAB ASC, Yellow)". Is this a monoclonal anti-ASC? Also, the images show large and bright antibody aggregates (middle of the image, top left corner behind the "H", and a massive fluorescence in the bottom right of the image), indicating the presence of staining artifacts. Again, no counterstaining with isotype controls are shown.

Response: We apologize for the confusing jargon in Figure 1H. "MAB ASC" refers to the anti-ASC^{PYD} antibody (MAB/MY6745). We have corrected the antibody terminology in the legend. MAB/MY6745 is a monoclonal antibody generated by Mabyon that is highly reactive to both human and murine ASC. This antibody was generated to 1) perform an immunotherapy *in vivo* study and to 2) be used as alternative specific antibody in addition to AL177 to show co-localization of SAA and ASC in a human AA patient using STED superresolution microscopy. MAB/MY6745 is a rabbit monoclonal anti-ASC antibody targeting the pyrin domain (PYD) from which the rabbit Fcγ domain was replaced with that of a mouse IgG_{2a} domain to avoid xenogeneic anti-drug responses in recipients and to improve its effector functions *in vivo*. To examine possible staining artefacts which can occur with Formalin-Fixed Paraffin-Embedded (FFPE) human tissues, we assessed the specificity of a variety of anti-ASC antibodies (Fig. S1). Our data presented in Fig. S1 show that the monoclonal anti-ASC antibody binds specifically. It is conceivable that AL177 and MAB/MY6745 target different epitopes of ASC, resulting in different staining patterns. An isotype control, included in Fig. S1, was used to test the specificity of the secondary antibodies, and did

not show any nonspecific staining. We have adapted and added this to the text body and figure legend accordingly.

Comment 3: Overall, although I don't dispute the possibility that ASC would co-localize with SAA deposits, I don't think the data presented can safely sustain that claim. I would, therefore, suggest that alternative methods to be employed to substantiate these conclusions: Supposedly, would it be possible to immuno-precipitate (IP) amyloid SAA and assess ASC via western blotting? As well as IP ASC and detect SAA? Or use DSS-crosslinking to find ASC oligomers in tissue areas rich in SAA?

Response: In addition to assessing co-localization by means of STED superresolution microscopy (**Fig. 1**), we also employed LIP-MS with various forms of ASC (monomeric and ASC specks) and identified a previously unrecognized biophysical interaction of SAA and the ASC PYD domain (**Fig. 2C-F**). As an orthogonal line of evidence, we provided kinetic data showing that SAA aggregation is enhanced in the presence of ASC specks (**Fig. 2A-B**). We feel that these results are reasonably convincing, but we agree that co-localization is almost invariably an aspirational finding, and even superresolution microscopy cannot fully exclude the presence artifacts (nor can, in fairness, co-immunoprecipitation, which must often rely on overexpression). A sentence acknowledging this limitation was added to the Discussion.

Comment 4: For example, it would be reasonable to quantify the results in Figure 3G and providing clarification regarding the controls in the figure legend. Though there is significantly less SAA in spleen homogenates from Asc^{-/-}, there also seems to be the case for b-actin in Fig 3G. Moreover, in the figure legend the authors state: "...Spleen homogenate from untreated (-ctrl) and AA⁺ (+ctrl) C57BL/6 wt mice from an independent experiment served as negative and positive control, respectively." I don't know what the authors mean with that. Is this a montage, or samples from different experiments were run together in one blot? And if so, for what reason? This is confusing and should be clarified.

Response: We reworded the figure legend to provide clarity about the technical assay controls and adjusted the labels in **Fig. 3E** accordingly: To ascertain SAA antibody functionality, mouse spleen homogenate from independently obtained and Congo red-confirmed AA⁺ tissue served as positive, whereas non-induced (AA⁻) spleen tissue served as negative technical controls. (**Fig 3E**). We decided to show the two (positive/AA⁺ and negative/AA⁻) technical controls in **Fig. 3E**.

Comment 5: Furthermore, in the Abstract, a slight rephrasing is suggested to accurately describe ASC specks as molecular aggregates formed inside cells, which are subsequently released into the extracellular space.

Response: We thank the referee for bringing this to our attention. We rephrased the abstract accordingly.

Comment 6: Lastly, enhancing the text size in figures, particularly in Fig 3, is advised to improve legibility and overall clarity.

Response: The legibility and style of main **Fig. 3** text sizes has been changed and additional figure formatting has been performed.

Referee 2

General: The manuscript by Losa et al., investigates whether ASC is involved in serum AA amyloidosis. The authors report that ASC colocalizes with SAA in human AA amyloidosis and that purified ASC specks accelerate SAA fibril formation in vitro. In addition, splenic AA amyloid was decreased in *Pycard*^{-/-} mice compared to *Pycard*^{+/+} mice and that treatment with anti-ASC antibodies decreased amyloid loads in *Pycard*^{+/+} mice. Lastly, they analyzed serum of 19,334 patients to show that the prevalence of anti-ASC antibodies did not correlate with any specific disease. The authors conclude that ASC to play a role in extraneural proteinopathies of humans and experimental animals and suggest that anti-ASC immunotherapy may contribute to resolving such diseases. The findings in the study are novel and demonstrate a new role for ASC in aggregation proteinopathies. However, there are number of issues that need to be addressed before acceptance for publication.

Significance: The findings in the study are novel and demonstrate a new role for ASC in aggregation proteinopathies. This study reports a crucial role for ASC in SAA interaction and recruitment, SAA serum level modulation, SAA fibril formation acceleration, and controlling the extent of inflammation associated amyloidosis with respect to AA amyloid deposition

Comment 1: Figure 3 E depicts Western blots of monomeric SAA in spleen of *Pycard*^{+/+} and *Pycard*^{-/-} mice. The authors should include immunoblots depicting the levels of ASC in these tissues and to demonstrate that the *Pycard*^{-/-} mice lack ASC.

Response: We did not perform ASC immunoblots for *Pycard*^{-/-} and *Pycard*^{+/+} mice since the absence of the ASC protein in this well-established mouse line has been demonstrated in several key publications, including under inflammation conditions (right side of the figure below, from Mariathasan et al., Nature, 2014). However, we show ASC IHC of *Pycard*^{+/+} and *Pycard*^{-/-} AA⁺ mice on spleen, confirming the absence of an ASC signal in *Pycard*^{-/-} mice and its presence in the *Pycard*^{+/+} (Fig. 3F). Moreover, our genotyping data confirmed the presence and absence of the *Pycard* gene in *Pycard*^{+/+} and *Pycard*^{-/-} AA⁺ mice.

Original *Pycard* knockout mice generation by Mariathasan et al.

Comment 2: Fig. 3B shows that at 96 hours after injection there was no difference in SAA serum concentration. How do the authors explain this drop in SAA serum concentration? No explanation is provided.

Response: Acute-phase response peaks at 24 hours after injury (i.e., Kushner I, 1982; Gabay et Kushner, 1999; Gitlin et Colten, 1987, *Calif.: Academic Press, 1987:123-53*). Beyond 24 hours, acute phase proteins decay over time mirroring the process of tissue integrity restoration and the clearance of the insulting stimuli. This is in line with our data, where the inflammatory injury was induced by subcutaneous AgNO₃ injection, resulting in a non-statistical serum SAA difference between the *Pycard*^{+/+} and *Pycard*^{-/-} experimental mice at 96 hours post AgNO₃ injection. In addition, the majority of SAA in *Pycard*^{+/+} mice was incorporated into amyloid deposit. As suggested by the reviewer we have included this explanation/references into the revised manuscript.

Comment 3: Figure 4 shows anti-ASC administration reduces amyloid load. The immunoblot in Figure 4C does not represent the quantification of the blot. In fact, there are only 3 samples per treatment group whereas the quantification shows 5-6 animals per group.

Response: We have performed two independent immunoblots at the same time to perform technical replicates (duplicates). As pointed out by the reviewer, this resulted in 6 samples and data points that were visualized and analyzed in main Fig. 4C. To avoid duplicating data, overloading the main figures with technical replicates, we opted to show only one representative immunoblot in the main Fig. 4C. The other blots are shown in the supplementary figures Fig. S13A and Fig. S13B for full transparency.

Comment 4: Additionally, the authors have not shown that the drug penetrates the target tissue and how much drug is present in spleen to provide a therapeutic effect. What is the half-life of the drug? These parameters are critical to assess the MOA of the anti-ASC used in these studies.

Response: To assess the pharmacokinetics of the anti-ASC antibody, we determined its titers in serum by ELISA at various time points up to 96 hpi after the first injection. The anti-ASC antibody serum levels peaked at 24 hpi and declined to about half maximal serum concentration levels at 96 hpi. This serum half-life, for the injected concentration, is in the range of reported kinetic parameters of engineered monoclonal antibodies (e.g., Unverdorben et al., MABs, 2016; Foss et al., Nat Comm, 2024) (Fig. 4B). Because of the high permeability of splenic red pulp vasculatures, and because of the absence of any selectively permeable barrier, efficacious imbibement of the splenic extracellular space can be plausibly expected. Theoretically, one could perfuse mice intracardially with PBS and then measure antibody in tissue. Such measurements can work relatively well in the brain, which possesses a highly impermeable barrier. However, here we would find it difficult to convince ourselves that such measurements would not be contaminated by residual blood in splenic capillaries that may be difficult to clean up through perfusion. Therefore, we did not measure the antibody levels in the spleen.

Comment 5: The authors should expand the discussion section to include the work of other groups that have successfully employed anti-ASC antibodies. For example, PMID: 35793783, PMID: 32366256

Response: We thank the referee for pointing out that literature. We extended the discussion section accordingly and added these important references into the discussion.

Comment 6: Methods: The authors provide the number of animals employed in the Supplemental Tables 5 and 7. These numbers should be provided in the methods section or in the Figure legends. Additionally, how many replicates were performed for the data in Figure 2?

Response: As suggested by the reviewer we now provide the number of animals in the figure legends of main Fig. 2 and Fig. 3 in addition to those in Table 5 and Supp Table 7 to enhance clarity.

Referee 3

General: The manuscript by Losa et al. explores the co-aggregation of ASC with serum amyloid A (SAA) in vivo and in mouse models. It posits that, similar to Amyloid beta, SAA is cross-seeded by ASC foci both in vitro and in vivo. This review only addresses the co-localization and in vitro cross seeding data (Figs. 1 and 2A, B), not the mouse experiments or mass spectrometry data.

The manuscript first shows co-deposition of ASC with SAA amyloid. SAA was stained both with Congo red and ThS, both standard dyes for amyloid staining. Figure S2 shows CR birefringence, the hallmark of amyloid deposits. The authors then move to demonstrate co-localization of SAA and ASC in confocal and STED immuno-fluorescence microscopy.

Significance: The discovery of the role of ASC in Alzheimer's disease generated an exciting new hypothesis to the etiology of sporadic AD, for which the cause is unknown. The current manuscript finds that ASC may also play a role in AA amyloidosis, which is a significant finding.

Comment 1: Confocal images C-E show overlapping staining of markers for both SAA and ASC. Similarly, STED images show co-aggregation of ASC and SAA in amyloidosis patients. However, since confocal images F and G seem to show overlapping staining of the yellow and magenta channels as well, a careful quantitative analysis of the data is needed. Quantify co-localization (Pearson coefficient) in confocal and STED images. STED images from control patients are missing and need to be included.

Response: AA amyloidosis is a relatively rare disease, and tissue samples thereof are even rarer. We only had access to the samples of one patient in both control and SAA groups. This limitation prevented us from conducting quantitative analyses. Rather than looking at the Pearson – or, possibly better, Spearman – correlation coefficient, we opted for an unbiased method of correlation in which we reconstructed the picture using 3D surface rendering with the Imaris software (see Fig. 1). From this reconstruction, we exported the barycenter of each surface on a 3D plot for both SAA and ASC markers (see Fig. S2B-C). Each point represents the center of a surface, while the box plots on the sides represent the distribution of the markers in space, demonstrating the overlap of the markers for ASC and SAA. We also understand the suggestion to conduct STED imaging on control samples to show the absence of co-aggregation. However, we could not be sure of which region to capture and how to decide on the focus, as we did not detect strong signal from confocal images of the control sample. Imaging blindly would almost necessarily lead to irrelevant imaging and aberrant comparison. We do not claim any quantitative data out of these images; however, we report an observation. Quantitative and mechanistic co-aggregation data are presented in Fig. 2 using LiP-MS.

Comment 2: The authors then move on to demonstrate that ASC foci can cross-seed SAA amyloid

Full Revision

formation *in vitro*, by recording SAA aggregation kinetics in the presence and absence of ASC foci. Curves recorded in the presence of ASC foci have accelerated kinetics as shown by a decrease in the time to reach half-maximal fluorescence ($t_{1/2}$). However, these data (Fig 2A, B) are not very clean. Only three data points out of five curves shown in panel A. are presented in the fitting of the control (yellow) aggregation kinetics in panel B. Why was this done? Panel B shows a significant difference between the control and the kinetics seeded with ASC specks. It looks doubtful that the results are still statistically significant if these data are included, so their exclusion impacts the overall conclusion of the paper. The significance of the cross-seeding results needs to be substantiated experimentally.

Response: The *in vitro* SAA aggregation assay was performed under established conditions (Claus S et al., EMBO Rep 2017) and the resulting data was processed using the AmyloFit software from the Knowles lab in Cambridge, UK (Meisl G et al., Nat Protoc 2016). The AmyloFit technology uses global fitting resulting in high-accuracy kinetics. Given the software algorithm, only curves that show a sigmoidal ThT fluorescence signal over time can be fitted. Therefore, replicates that do not show aggregation (characteristic ThT signal) over time cannot be fitted. As a result, only three out of six curves could be fitted resulting in three $t_{1/2}$. Conversely, in the presence of ASC specks, all six replicates aggregated in a dose-dependent manner, and could be fitted perfectly, yielding six $t_{1/2}$ values. Thus, all available data points are plotted and used for statistical analysis. Moreover, the fact that in presence of ASC specks all SAA replicates aggregated/converted successfully in a dose-dependent manner (whereas in the SAA-only condition some replicates do not aggregate) further underscores the pivotal role of ASC specks in SAA seeding, conversion, and aggregation enhancement.

18th Jun 2024

Dear Adriano,

Thank you for the submission of your revised manuscript to EMBO Molecular Medicine. We have now received the enclosed report from the three referees, who agreed to re-assess it. As you will see, the referees are supportive, and I am pleased to inform you that we will be able to accept your manuscript pending the following amendments:

1. Please provide up to five keywords.

2. Figures and tables:

- Main figures should be removed from the manuscript text and uploaded as individual, high resolution figure files, and their legends should be compiled in the manuscript text, after the References.
- The supplementary figures, legends and tables should be removed from the manuscript and compiled in a PDF file labelled 'Appendix'. The appendix needs a table of contents with page numbers, and the nomenclature should be corrected to "Appendix Figure S1" etc. and "Appendix Table S1". Callouts in the manuscript need to be updated accordingly.
- Figure callouts are missing for Table S1 and S3.

3. Remove "data not shown" (in the legend of Figure S7). As per our guidelines on "Unpublished Data" the journal does not permit citation of "data not shown". All data referred to in the paper should be displayed in the main or Expanded View figures/Appendix Figures.

4. The references need to be formatted according to the EMBO Molecular Medicine reference style. Citations should be listed in alphabetical order and list 10 co-authors of a paper before adding et al. DOIs of all published papers need to be removed.

5. Remove the Author contribution section from the manuscript file.

6. Please use the heading "Disclosure statement and competing interests" for the declaration of interest statement. Please add "Adriano Aguzzi is a member of the Advisory Editorial Board of EMBO Molecular Medicine. This has no bearing on the editorial consideration of this article for publication."

7. The provided funding information should be consistent between the online submission system and the manuscript text. PHN driver grant (2017DRI1), NOMIS Foundation, Schwyzer Winiker Stiftung, and the Baugarten Stiftung (coordinated by the USZ Foundation-USZF27101) are mentioned in the Acknowledgement but not in the submission system. Please fix it.

8. Data and Material statement: Please use the heading "Data availability". Before submitting your revision, primary datasets (RNA-seq and mass spectrometry data) produced in this study need to be deposited in an appropriate public database (see <https://www.embopress.org/page/journal/17574684/authorguide#dataavailability>). The accession numbers and database should be listed in a formal "Data Availability" section (placed after Materials & Method) that follows the model below (see also <https://www.embopress.org/page/journal/17574684/authorguide#dataavailability>). Please note that the Data Availability Section is restricted to new primary data that are part of this study.

Data availability

9. At EMBO Press we ask authors to provide source data for the main figures. Our source data coordinator will contact you to discuss which figure panels we would need source data for and will also provide you with helpful tips on how to upload and organize the files.

10. For more information: There is space at the end of each article to list relevant web links for further consultation by our readers. Could you identify some relevant ones and provide such information as well? Some examples are patient associations, relevant databases, OMIM/proteins/genes links, author's websites, etc...

11. The Paper Explained: EMBO Molecular Medicine articles are accompanied by a summary of the articles to emphasize the major findings in the paper and their medical implications for the non-specialist reader. Please provide a draft summary of your article highlighting

This may be edited to ensure that readers understand the significance and context of the research. Please refer to any of our

published articles for an example.

12. Every published paper now includes a 'Synopsis' to further enhance discoverability. Synopses are displayed on the journal webpage and are freely accessible to all readers. They include a short stand first (maximum of 300 characters, including space) as well as 2-5 one sentence bullet points that summarise the paper. Please write the bullet points to summarise the key NEW findings. They should be designed to be complementary to the abstract - i.e. not repeat the same text. We encourage inclusion of key acronyms and quantitative information (maximum of 30 words / bullet point). Please use the passive voice. Please attach these in a separate file or send them by email, we will incorporate them accordingly.

Please also suggest a striking image or visual abstract to illustrate your article. If you do please provide a jpeg file 550 px-wide x 400-px high.

13. When you resubmit your manuscript, please download our CHECKLIST (<https://www.embopress.org/pb-assets/embo-site/EMBO%20Press%20Author%20Checklist-1642513524327.xlsx>) and include the completed form in your submission. *Please note* that the Author Checklist will be published alongside the paper as part of the transparent process (<https://www.embopress.org/page/journal/17444292/authorguide#transparentprocess>).

14. Please address the following requests from our data editor:

- Figure legends:

1. Please note that the exact p values are not provided in the legends of figures 2b; 3b, d-e, h; 4c, f, h.
2. Please note that in figures 4c, f-h; there is a mismatch between the annotated p values in the figure legend and the annotated p values in the figure file that should be corrected.
3. Please note that the box plots need to be defined in terms of minima, maxima, and whiskers in the legend of figure 2b.
4. Please note that the box plots need to be defined in terms of minima, maxima, centre, bounds of box and whiskers, and percentile in the legend of figure 5c.
5. Please note that information related to n is missing in the legend of figure 4b.
6. Please note that the error bars are not defined in the legend of figure 4b.

Please submit your revised manuscript within two weeks. I look forward to seeing a revised form of your manuscript as soon as possible.

Best wishes,
Jingyi

Jingyi Hou
Editor
EMBO Molecular Medicine

*** Instructions to submit your revised manuscript ***

- 1) a .docx formatted version of the manuscript text (including Figure legends and tables)

2) Separate figure files*

3) supplemental information as Expanded View and/or Appendix. Please carefully check the authors guidelines for formatting Expanded view and Appendix figures and tables at <https://www.embopress.org/page/journal/17574684/authorguide#expandedview>

4) a letter INCLUDING the reviewer's reports and your detailed responses to their comments (as Word file).

5) The paper explained: EMBO Molecular Medicine articles are accompanied by a summary of the articles to emphasize the major findings in the paper and their medical implications for the non-specialist reader. Please provide a draft summary of your article highlighting

This may be edited to ensure that readers understand the significance and context of the research.

Please refer to any of our published articles for an example.

6) For more information: There is space at the end of each article to list relevant web links for further consultation by our readers. Could you identify some relevant ones and provide such information as well? Some examples are patient associations, relevant databases, OMIM/proteins/genes links, author's websites, etc...

7) Author contributions: the contribution of every author must be detailed in a separate section.

8) EMBO Molecular Medicine now requires a complete author checklist

(<https://www.embopress.org/page/journal/17574684/authorguide>) to be submitted with all revised manuscripts. Please use the checklist as guideline for the sort of information we need WITHIN the manuscript. The checklist should only be filled with page numbers where the information can be found. This is particularly important for animal reporting, antibody dilutions (missing) and exact values and n that should be indicated instead of a range.

9) Every published paper now includes a 'Synopsis' to further enhance discoverability. Synopses are displayed on the journal webpage and are freely accessible to all readers. They include a short stand first (maximum of 300 characters, including space) as well as 2-5 one sentence bullet points that summarise the paper. Please write the bullet points to summarise the key NEW findings. They should be designed to be complementary to the abstract - i.e. not repeat the same text. We encourage inclusion of key acronyms and quantitative information (maximum of 30 words / bullet point). Please use the passive voice. Please attach these in a separate file or send them by email, we will incorporate them accordingly.

You are also welcome to suggest a striking image or visual abstract to illustrate your article. If you do please provide a jpeg file 550 px-wide x 300-800px high.

10) A Conflict of Interest statement should be provided in the main text

11) Please note that we now mandate that all corresponding authors list an ORCID digital identifier. This takes <90 seconds to complete. We encourage all authors to supply an ORCID identifier, which will be linked to their name for unambiguous name identification.

Currently, our records indicate that the ORCID for your account is 0000-0002-0344-6708.

Link Not Available

Photos 400-800 DPI

*Additional important information regarding figures and illustrations can be found at

<https://bit.ly/EMBOPressFigurePreparationGuideline>. See also figure legend preparation guidelines:
<https://www.embopress.org/page/journal/17574684/authorguide#figureformat>

**** Reviewer's comments ****

Referee #1 (Comments on Novelty/Model System for Author):

As stated before the initial review process, my comments can only cover the technical aspects of microscopy and in vitro aggregation experiments, not the MS or mouse models. I can therefore neither evaluate the novelty, medical impact or the adequacy of the model system.

Referee #1 (Remarks for Author):

The authors have comprehensively addressed my previous concerns with regard to co-localization analysis in confocal and STED imaging.

The authors did not substantially address my second comment with regard to seeding analysis (Fig 2A) other than stating that the experiment hadn't run for long enough to have fittable curves for all the data. However, the lack of quantitative fitting of part of the in Fig 2A does not affect the main conclusions of the manuscript.

Referee #2 (Comments on Novelty/Model System for Author):

The study has improved from the original draft and the claims are supported by the data provided. I'm also glad that the authors indicated the limitations of their study in a well-balanced manner.

Referee #2 (Remarks for Author):

The authors have successfully addressed all my comments. I restate my words on the study's impact and how it advances the knowledge of IL-1-independent inflammatory functions of inflammasomes. This is a well-conducted and well-written study, and I endorse its publication.

Referee #3 (Comments on Novelty/Model System for Author):

The manuscript demonstrates that ASC controls inflammation associated AA amyloidosis, a systemic disease caused by the aggregation and peripheral deposition of the acute-phase reactant serum amyloid A (SAA).

Referee #3 (Remarks for Author):

The authors have adequately addressed all concerns raised.

Rev_Com_number: RC-2023-02235

New_manu_number: EMM-2024-20060-T

Corr_author: Aguzzi

Title: The ASC inflammasome adapter governs the extent of peripheral SAA-derived protein aggregate deposition in inflammation-induced amyloidosis

The authors addressed the minor editorial issues.

10th Jul 2024

Dear Adriano,

We are pleased to inform you that your manuscript is accepted for publication and is now being sent to our publisher to be included in the next available issue of EMBO Molecular Medicine.

Kind regards,
Jingyi

Jingyi Hou
Editor
EMBO Molecular Medicine

Rev_Com_number: RC-2023-02235

New_manu_number: EMM-2024-20060-V2

Corr_author: Aguzzi

Title: The ASC inflammasome adapter governs SAA-derived protein aggregation in inflammatory amyloidosis